# ATG9A regulates the dissociation of recycling endosomes from microtubules to form liquid influenza A virus inclusions

Sílvia Vale-Costa[1], Temitope Akhigbe Etibor[1], Daniela Brás[1], Ana Laura Sousa[2], Mariana Ferreira[3], Gabriel G. Martins[3], Victor Hugo Mello[4], Maria João Amorim[1,5]*

1 Cell Biology of Viral Infection Lab (CBV), Instituto Gulbenkian de Ciência (IGC)—Fundação Calouste Gulbenkian, Oeiras, Portugal, 2 Electron Microscopy Facility (EMF), Instituto Gulbenkian de Ciência (IGC)—Fundação Calouste Gulbenkian, Oeiras, Portugal, 3 Advanced Imaging Facility (AIF), Instituto Gulbenkian de Ciência (IGC)—Fundação Calouste Gulbenkian, Oeiras, Portugal, 4 Living Physics, Instituto Gulbenkian de Ciência (IGC)—Fundação Calouste Gulbenkian, Oeiras, Portugal, 5 Cell Biology of Viral Infection Lab (CBV), Católica Biomedical Research Centre (CBR), Católica Medical School—Universidade Católica Portuguesa, Lisboa, Portugal

* mjamorim@igc.gulbenkian.pt; mjamorim@ucp.pt

**Data Availability Statement:** All relevant data are within the paper and its Supporting Information files. All metadata files are available from the Zenodo database: https://doi.org/10.5281/zenodo.

## Abstract

It is now established that many viruses that threaten public health establish condensates via phase transitions to complete their lifecycles, and knowledge on such processes may offer new strategies for antiviral therapy. In the case of influenza A virus (IAV), liquid condensates known as viral inclusions, concentrate the 8 distinct viral ribonucleoproteins (vRNPs) that form IAV genome and are viewed as sites dedicated to the assembly of the 8-partite genomic complex. Despite not being delimited by host membranes, IAV liquid inclusions accumulate host membranes inside as a result of vRNP binding to the recycling endocytic marker Rab11a, a driver of the biogenesis of these structures. We lack molecular understanding on how Rab11a-recycling endosomes condensate specifically near the endoplasmic reticulum (ER) exit sites upon IAV infection. We show here that liquid viral inclusions interact with the ER to fuse, divide, and slide. We uncover that, contrary to previous indications, the reported reduction in recycling endocytic activity is a regulated process rather than a competition for cellular resources involving a novel role for the host factor ATG9A. In infection, ATG9A mediates the removal of Rab11a-recycling endosomes carrying vRNPs from microtubules. We observe that the recycling endocytic usage of microtubules is rescued when ATG9A is depleted, which prevents condensation of Rab11a endosomes near the ER. The failure to produce viral inclusions accumulates vRNPs in the cytosol and reduces genome assembly and the release of infectious virions. We propose that the ER supports the dynamics of liquid IAV inclusions, with ATG9A facilitating their formation. This work advances our understanding on how epidemic and pandemic influenza genomes are formed. It also reveals the plasticity of recycling endosomes to undergo condensation in response to infection, disclosing new roles for ATG9A beyond its classical involvement in autophagy.

8183202 (for main figures) and in https://doi.org/
10.5281/zenodo.8190137 (for supplementary
figures).

**Funding:** This project has received funding from
the European Research Council (ERC) under the
European Union's Horizon 2020 research and
innovation programme (grant agreement No.
101001521 to MJA) and by National Funds from
the Portuguese Fundação para a Ciência e a
Tecnologia (2022.02716.PTDC_EXPL WP1 to
SVC). This work was also supported by the
Instituto Gulbenkian de Ciência Advanced Imaging
Facility, which is funded by PPBI-POCI-01-0145-
FEDER-022122 (Lisboa 2020/FEDER/FCT; Portugal
to GB), and by the Electron Microscopy Facility and
Flow Cytometry Facility, which are funded by
Fundação Calouste Gulbenkian (Portugal, Lisbon)
to the facilities. Salary support from FCT: TAE, DB,
VM are funded by PhD fellowships (PD/BD/
128436/2017, PD/BD/148391/2019 and UI/BD/
152254/2021, respectively) and SVC by D.L. 57.
The funders had no role in study design, data
collection and analysis, decision to publish, or
preparation of the manuscript.

**Competing interests:** The authors have declared
that no competing interests exist.

**Abbreviations:** ATG9A, autophagy related gene 9A;
DMEM, Dulbecco's Modified Eagle's Medium; ER,
endoplasmic reticulum; ERES, ER exit site; ET,
electron tomography; FBS, fetal bovine serum; FIP,
family interacting protein; GFP, green fluorescent
protein; HA, hemagglutinin; IAV, influenza A virus;
LCIS, live cell imaging solution; MDCK, Madin-
Darby canine kidney; MOI, multiplicity of infection;
MSD, mean squared displacement; M2, matrix
protein 2; NA, neuraminidase; NP, nucleoprotein;
PB, phosphate buffer; PFA, paraformaldehyde;
PFU, plaque-forming unit; RT-qPCR, quantitative
reverse transcription PCR; SEM, standard error of
the mean; TEM, transmission electron microscopy;
Tf, transferrin; TGN, *trans*-Golgi network; TIS, TPA-
induced sequence; vRNP, viral ribonucleoprotein;
WT, wild-type; 2D, 2-dimensional; 3D, 3-
dimensional.

## Introduction

Influenza A virus (IAV) is a major causative agent of yearly flu epidemics responsible for high
mortality and morbidity, despite worldwide surveillance of circulating viruses, yearly vaccina-
tion programs, and availability of antivirals. This zoonotic virus has presented occasional host-
species jumps from other animals (birds, pigs) that have led to pandemics of serious conse-
quences (reviewed in [1]). Underlying factors contributing to the perpetuation of IAV circula-
tion in humans (and other animals) combine viral mutation rate and genomic mixing between
different IAV strains. Genomic mixing accelerates viral evolution and is feasible as the IAV
genome is segmented, composed by 8 distinct RNA segments arranged into viral ribonucleo-
proteins (vRNPs). Despite the advantage for fast viral evolution, genomic segmentation poses
an interesting challenge for genome assembly, as it is known that most IAV virions contain
exactly 8 vRNPs and one of each kind (reviewed in [2]). Decades of seminal research have con-
vincingly demonstrated that IAV genome assembly is a selective process, involving interseg-
ment RNA–RNA interactions (reviewed in [3]). However, to date, the molecular mechanism
governing the assembly of influenza genomes remains unclear.

We have recently proposed an appealing model to explain IAV genome assembly [4],
which involves the formation of biomolecular condensates designated viral inclusions. We
found that IAV viral inclusions share properties with bona fide liquid condensates formed by
liquid–liquid phase separation–based processes [4,5]. They are not delimited by a membrane,
are highly dynamic, react to stimuli, and internally rearrange [4,5]. Interestingly, despite not
being delimited by membranes, IAV inclusions result from the accumulation of Rab11a recy-
cling endosomes interacting with the different vRNP types, which are embedded as part of
condensates [4,6]. In our model, the liquid-like character results from a network of weakly
interacting vRNPs that bridge multiple cognate vRNP-Rab11a units on flexible membranes
resulting presumably in percolation and condensation [5], which is currently being validated
in our lab using in vitro reconstitution systems. More than just a confined space wherein IAV
genome assembly may be efficiently orchestrated, viral inclusions with liquid properties con-
stitute a change in paradigm that offer new hypotheses to test how IAV genomic complexes
form. In fact, the flexibility of movement within liquid structures combined with critical recent
advances in understanding the rules governing the formation of cellular biomolecular conden-
sates [5] raises the possibility that complete genomes may have different affinities for conden-
sates. Viral inclusions with liquid properties are important for IAV replication. This is
supported by evidence that abrogating the formation [4,7–11] or forcing viral inclusions to
transition from a liquid into a hardened state efficiently blocks viral production in cellular and
animal infection models [5]. It also illustrates that modulating the material state of viral inclu-
sions could become an innovative strategy to control influenza infections.

The only confirmed cellular driver of viral inclusion formation is Rab11a, which has a role
well established in recycling slow cargo to the plasma membrane in uninfected cells, binding
directly to Rab11 family interacting proteins (FIPs) that, in turn, recruit molecular motors
[12]. During IAV infection, the initial view that Rab11a transported vRNPs to the plasma
membrane [7–10,13–15] was challenged by reports demonstrating that Rab11a-mediated recy-
cling was hindered [16,17] resulting in the formation of liquid viral inclusions [4,6]. Binding
of vRNPs and FIPs at the same domains in Rab11 [6,18] suggested that vRNPs outcompeted
FIPs and molecular motors impairing cytoskeletal-based transport [6,9]. Interestingly, recent
research indicates FIP binding to Rab11a remains unaffected during infection, hinting at a reg-
ulated process involving dynein in viral inclusion formation [16]. This, coupled with Rab11's
association with modified endoplasmic reticulum (ER) [30] and proximity of IAV liquid inclu-
sions to ER exit sites (ERES) [4], strongly suggests a connection between recycling endosomes,

the ER, and IAV genome assembly. However, which cellular factors regulate the biogenesis and dynamics of viral inclusions near the ER are yet to be defined.

Accumulating evidence shows that membrane-bound organelles and liquid biomolecular condensates may intimately interact in physiological contexts (reviewed in [19,20]). In line with this, the ER has occupied a central role [21–23]. The ER has critical and numerous roles in the cell, from protein and lipid synthesis, to carbohydrate metabolism, and calcium storage and signaling [24]. It has an expansive membrane able to easily rearrange and to connect with other intracellular organelles in response to specific stimuli [24]. Interestingly, the ER was shown to act as a platform for the phase separation of Tiger and Whi3 ribonucleoproteins, TIS (TPA-induced sequence) granules, Sec bodies, and autophagosome nucleation sites (reviewed in [19,20]), and it was shown to regulate the fission of liquid ribonucleoprotein granules to maintain their size [23]. Further examples on the interplay between membrane-bound organelles and biomolecular condensates include the demonstration that phase separated synaptic vesicles form as a mechanism for ready deployment for neurotransmission release [25].

In this study, we sought to better define the interplay between the ER and IAV liquid inclusions. We observed that the ER supports viral inclusion fusion, fission, and sliding movements as reported for other RNP condensates [21]. From an siRNA screen of host factors involved in early steps of autophagy, we identified ATG9A (autophagy related gene 9A) as a host factor that impacted IAV liquid inclusion biogenesis and viral replication cycle. We found that ATG9A regulated trafficking of liquid viral inclusions between the ER and microtubules, removing recycling endosomes from microtubules and leading to their condensation close to ERES. ATG9A was initially identified as a core member of the autophagic machinery, mechanistically flipping phospholipids between the 2 membrane leaflets of the autophagosomal membrane to promote its growth [26]. However, we find that key initial players in autophagy (ULK1/2, TBC1D14, or ATG2A) did not regulate ER–microtubule trafficking, even though ULK2 affected virion production, suggesting that this function of ATG9A is novel. In fact, ATG9A was reported to display other roles unrelated to autophagy, including plasma membrane repair [27], lipid mobilization between organelles [28], and regulation of innate immunity [29]. Here, we show that ATG9A modulates liquid–liquid phase separation on or near the ER in mammalian cells. Interestingly, it was reported that ATG9A was able to modulate FIP200 phase separation adjacent to the ER during autophagy [22]. In this paper, we further contribute to understanding this mechanism by establishing a link between ATG9A and microtubules that has never been reported. It also contributes to how biomolecular condensates form by reprogramming preexisting pathways. The formation of numerous liquid condensates in the cell is initiated in response to specific stimuli. Therefore, our study has broader implications for biological systems by demonstrating the flexibility of unforeseen cellular machinery to change its function giving rise to biomolecular condensates.

## Results

### Rab11a-regulated recycling is impaired by IAV infection

We have recently shown that liquid viral inclusions, condensates enriched in Rab11a endosomes and vRNPs, develop in the vicinity of the ER subdomain ERES [4]. Hence, both Rab11a and the proteins associated with vRNPs (viral RNA polymerase subunits and nucleoprotein (NP)) can be used as a proxy to visualize viral inclusions. How Rab11a endosomes accumulate near the ER to form viral inclusions and how the recycling function is consequently affected during IAV infection is unclear. We hypothesize that, upon nuclear export, progeny vRNPs bind to Rab11a endosomes, which, together, are rerouted to the ER to form viral inclusions. As a consequence, Rab11a recycling capacity is expected to be impaired during IAV infection

(Fig 1A, steps 1 to 2). This hypothesis is supported by several pieces of evidence. First, given that the cytoplasmic content of vRNPs increases as infection progresses and that vRNPs bind Rab11a (via PB2 viral protein) [7,8,18], we have shown that vRNPs outcompete Rab11a adaptors/molecular motors for Rab11a binding [6]. Second, as a consequence of such competition, we have shown that transferrin (Tf) recycling is reduced throughout infection [6]. Third, another group has detected the presence of Rab11a and vRNPs close to membranes of a remodeled ER during IAV infection [30].

Here, we extended our previous studies [4,6] to gain mechanistic insight into the fate of Rab11a during IAV infection. Our aim was to demonstrate that IAV infection impairs Rab11a-regulated recycling and to show that Rab11a endosomes accumulate near the ER to form viral inclusions (Fig 1A, steps 1 to 2). The combined visual inspection of Rab11 distribution in mock and infected cells with transferrin (Tf) recycling assays allows determining the fate of Rab11a endosomes in infection. We used A549 lung epithelial cells expressing low levels of Rab11a wild-type (GFP-Rab11a WT$^{low}$) and dominant-negative (GFP-Rab11a DN$^{low}$) fused to green fluorescent protein (GFP) and infected or mock-infected with PR8 virus for 12 h. Cells expressing GFP-Rab11a WT$^{low}$ produce significantly more viruses (2.5 log) than GFP-Rab11a DN$^{low}$ at 12 h after infection (S1A Fig, mean plaque-forming units (PFUs).mL$^{-1}$ ± standard error of the mean (SEM): WT– 141,875 ± 65,599 versus DN– 360 ± 142). Besides that, GFP-Rab11a WT$^{low}$ cells are able to form large cytosolic puncta or viral inclusions, whereas GFP-Rab11a DN$^{low}$ cells are unable to mount these condensates near the ER (S1B Fig), as we have shown before [4,6]. These results indicate that both cell lines are adequate to analyze Rab11a-regulated recycling as well as Rab11a dynamics during IAV infection.

To test if Rab11a-regulated recycling is altered by IAV infection, we quantified by flow cytometry the recycling capacity of both cell lines infected or mock-infected with PR8 virus for 12 h (Fig 1B and 1C). Upon feeding with a Tf-Alexa647-fluorescent conjugate, a classical cargo protein shown to be recycled by Rab11a endosomes [31–33], cells were allowed to recycle Tf for 5, 10, and 15 min at 37°C. We observed that both cell types have a significantly decreased ability to recycle Tf upon infection, in comparison to the respective mock-infected cells (Fig 1B). The drop in Tf recycling (at 15 min) caused by infection in GFP-Rab11a WT$^{low}$ cells is 53.5%, whereas in GFP-Rab11a DN$^{low}$ cells, the reduction in recycling is more pronounced, being around 75.4% (Fig 1B). When both cell types were compared directly (at 15 min; Fig 1C), we observed a reduction in Tf recycling levels caused by infection (% mean Tf recycling ± SEM: WT Mock– 100.0 ± 0.0% versus WT PR8–47.7 ± 5.1%, DN Mock– 87.2 ± 2.3% versus DN PR8–27.1 ± 4.5%). Of note, both infected and mock-infected GFP-Rab11a DN$^{low}$ cells have a small decrease in Tf recycling relative to GFP-Rab11a WT$^{low}$ cells. This reduction slightly intensifies during infection (12.8% to 20.6%, respectively; Fig 1C). This points to Tf recycling being facilitated through redundant pathways (Rab4, Rab10, Rab11) [12] in uninfected cells, with a modest involvement of Rab11a. With PR8 infection, Tf recycling decreases significantly, and a modest increase on Rab11a dependence is observed.

Together, our results demonstrate that all cellular recycling is impaired during IAV infection. Moreover, the observation of enlarged cytosolic Rab11a puncta (corresponding to the liquid viral inclusions) near the ER agrees with previous reports and confirms that Rab11a pathway is severely affected in infected cells [4,6,7].

## Viral inclusions dynamically interact with the ER

Our previous results showed that Rab11a-regulated recycling function is impaired during IAV infection (Fig 1A–1C) and that Rab11a endosomes accumulate near the ER [4]. It is unknown why Rab11a endosomes are rerouted specifically to the ER during IAV infection. One

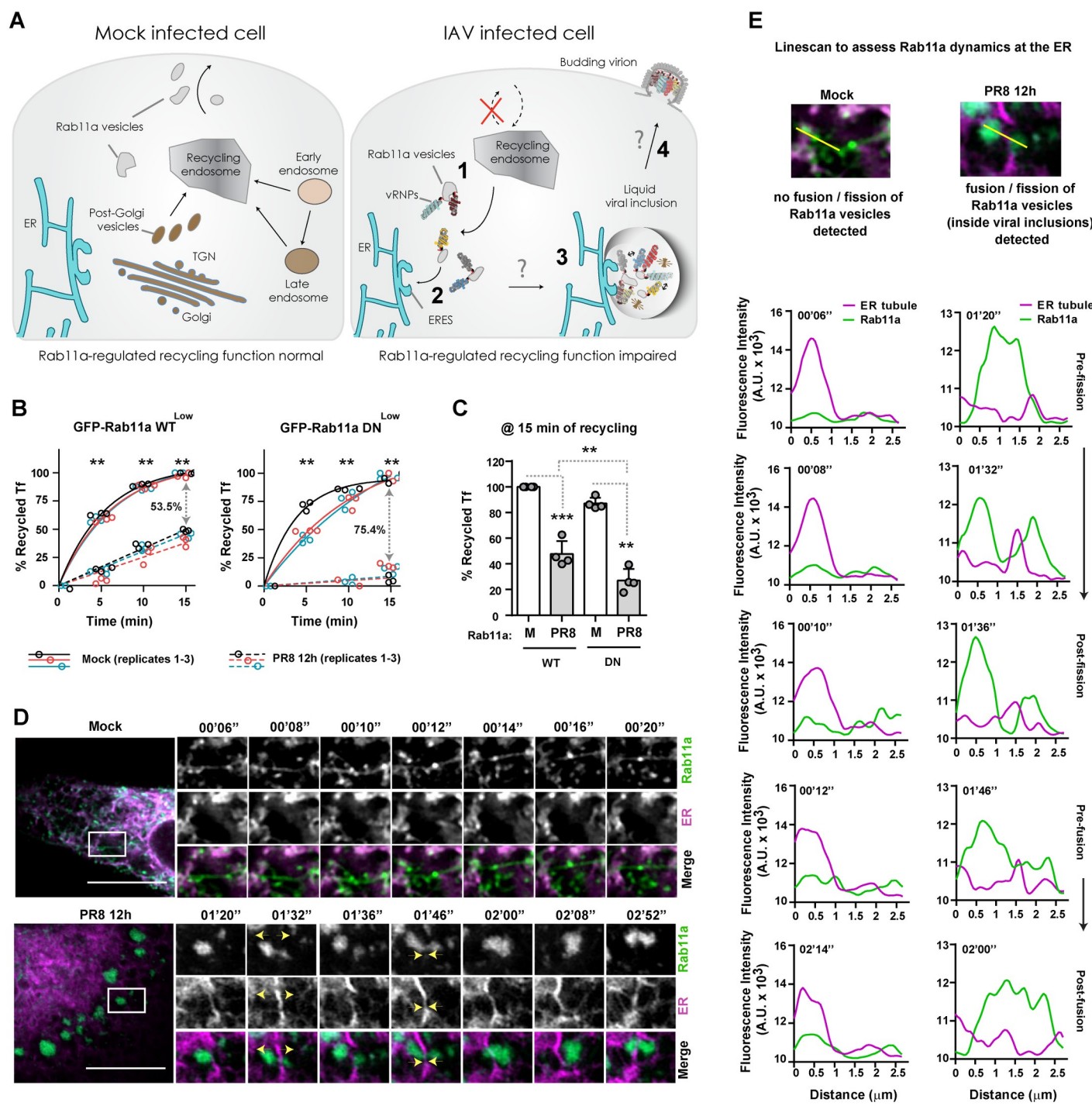

**Fig 1. IAV infection impairs Rab11a-regulated recycling and sustains the dynamics of viral inclusions. (A)** Schematic representation of Rab11a-regulated recycling in mock- and IAV-infected cells. In mock-infected cells, Rab11a endosomes are involved in recycling material from several organelles to the plasma membrane. Upon IAV infection, progeny vRNPs bind to Rab11a endosomes and start concentrating at ERES (steps 1–2) to form liquid viral inclusions by a mechanism ill-defined (step 3). The ER likely facilitates formation of viral inclusions to promote assembly of the 8-vRNP genome (step 3). How assembled genomes reach the plasma membrane is unknown (step 4). **(B)** Cells (GFP-Rab11a WT$^{low}$ and DN$^{low}$) were infected or mock-infected with PR8 virus for 12 h at an MOI of 3. The levels of Tf-Alexa647-fluorescent conjugates were quantified inside cells and at the cell surface by flow cytometry upon 5, 10, and 15 min of incubation at 37°C. Results were plotted as the percentage (%) of recycled Tf as a function of time of incubation. Values were normalized to the mock condition at 15 min of incubation. Three pooled independent experiments are shown. Statistical analysis was done by two-way ANOVA, followed by a Sidak's multiple comparisons test (**$p < 0.01$). **(C)** The levels of Tf were quantified inside cells and at the cell surface by flow cytometry at 15 min of incubation at 37°C. Results were plotted as the percentage (%) of recycled Tf as a function of cell type. Four pooled independent

experiments are shown. Statistical analysis was done by one-way ANOVA, followed by a Tukey's multiple comparisons test (**$p < 0.01$, ***$p < 0.001$). **(D)** Cells (GFP-Rab11a WT$^{low}$, green) were simultaneously transfected with a plasmid encoding mCherry tagged to the ER (magenta) and infected or mock-infected with PR8 virus for 12 h at an MOI of 10. Cells were imaged under time-lapse conditions at 12 h postinfection. Representative cells are shown on the left. The respective individual frames with single moving particles are shown in the small panels on the right. The yellow arrowheads highlight fusion/fission events of viral inclusions (green), as well as their interaction with the ER (magenta). Bar = 10 μm. Images were extracted from S1 and S2 Videos. **(E)** A linescan was drawn as indicated to assess Rab11a dynamics associated with the ER. The fluorescence intensity of ER tubules (magenta) and Rab11a endosomes or viral inclusions (green) at indicated times was plotted against the distance (in μm). Representative analysis was performed using images from **(D)**. Experiments were performed twice. For each condition, at least 10 cells were analyzed. All the values of individual and pooled experiments are provided in S1 Data File. ER, endoplasmic reticulum; ERES, ER exit site; IAV, influenza A virus; MOI, multiplicity of infection; Tf, transferrin; vRNP, viral ribonucleoprotein.

possibility suggested by other authors is that the ER is involved in trafficking vRNPs [30]. We propose here an additional step in which ER membranes are key sites for concentrating Rab11a endosomes and vRNPs, promoting the biogenesis, dynamics, and controlling the size of liquid viral inclusions. This is supported by the fact that liquid viral inclusions dissolve if vesicular cycling between the ER and Golgi is impaired [4]. Moreover, recent studies demonstrated that the ER acted as a platform for the phase separation of numerous biomolecular condensates (reviewed in [19]) and can regulate their size by promoting fission events [21].

In order to visualize dynamic interactions between Rab11a (used as a proxy for viral inclusions) and the ER, we performed live imaging of GFP-Rab11a WT$^{low}$ cells (green) transfected with a plasmid encoding mCherry tagged to the ER (magenta) and simultaneously infected or mock-infected with PR8 virus for 12 h. As expected, infected GFP-Rab11a WT$^{low}$ cells formed large and rounded viral inclusions that dynamically exchanged material (Fig 1D and S1 Video). We could detect Rab11a on- and off-contacts and sliding movements on the ER, as well as fission and fusion events supported by the ER (Fig 1D and 1E, yellow arrows), similar to those described for vRNPs [4]. In mock-infected cells, Rab11a presents as a tubulovesicular network and, although short-lived contacts between Rab11a and the ER can be occasionally detected, the majority of Rab11a does not localize at the ER and does not fuse or fissions (Fig 1D and 1E and S2 Video). A similar analysis was not performed in GFP-Rab11a DN$^{low}$ cells, as viral inclusions do not form in the absence of a functional Rab11a (S1B Fig and [4,6]).

Overall, our results suggest that the ER supports the dynamics of liquid viral inclusions. In uninfected cells, Rab11a endosomes are involved in recycling material from several organelles (early and late endosomes, and *trans*-Golgi network (TGN)) to the cell surface (Fig 1A) [12]. Upon IAV infection, progeny vRNPs bind to Rab11a endosomes [4,6–8,18] and start concentrating at ERES (Fig 1A, steps 1 to 2) [4] to form viral inclusions by a mechanism incompletely understood (Fig 1A, step 3). As a consequence, Rab11a-regulated recycling function is impaired in infection. Our data strongly suggest that the ER facilitates fusion/fission of liquid viral inclusions to likely promote the exchange of vRNPs and the assembly of the 8-vRNP genome (Fig 1A, step 3). How assembled genomes reach the budding sites at the plasma membrane is not yet known, but such a question is outside the scope of this study (Fig 1A, step 4).

## Viral inclusions contain single- and double-membrane vesicles inside and locate near the ER

To better understand the development of viral inclusions near the ER, we characterized their ultrastructure in GFP-Rab11a WT$^{low}$ cells infected or mock-infected with PR8 virus for 12 h. We have used these cell lines before to identify viral inclusions as cytosolic sites positive for Rab11a and vRNPs, using distinct light and electron imaging methodologies [4,6]. Our previous 2-dimensional (2D) ultrastructural analysis of viral inclusions, using correlative light and electron microscopy, revealed aggregates of double-membrane structures and single-membrane vesicles of heterogeneous sizes decorated with vRNPs [4,6], as schematically exemplified

(Fig 2A). Another group observed irregularly coated vesicles protruding from a dilated and tubulated ER [30], densely covered with vRNPs and Rab11a. In both studies, ultrastructural analysis was performed using chemical fixation and plastic sectioning, which can introduce artifacts and structural distortions.

To consolidate both observations and overcome these methodological limitations, we resolved the 3-dimensional (3D) organization of viral inclusions by high-pressure freezing/ freeze substitution and electron tomography (ET) transmission electron microscopy (TEM). For 3D model reconstruction, 4 serial tomograms of 120 nm each were stitched together (480 nm thickness in total; Fig 2B), of which 3 representative sections are shown (Fig 2C and 2D). The 3D model of an IAV inclusion revealed numerous single-membrane vesicles (smv, light green) of heterogeneous sizes clustered around a double-membrane structure (dmv, yellow) close to the ER (er, blue) in infected GFP-Rab11a WT$^{low}$ cells (Fig 2D and S3 and S4 Videos). We also detected the presence of ER dilations (*dark green; Fig 2D). In opposition, mock-infected cells had numerous single-membrane vesicles near the plasma membrane (pm, gray) or scattered in the cytoplasm, and double-membrane vesicles or dilated ER could not be found (Fig 2C and S5 and S6 Videos). The ultrastructural features of viral inclusions observed in infected GFP-Rab11a WT$^{low}$ cells were also present in A549 cells containing endogenous Rab11a levels. The similarity in ER alterations indicates that it is infection, and not Rab11a overexpression that changes the ER morphology, which leads to conclude that our observations are not an artifact caused by Rab11a overexpression (S2A Fig and S7, S8, S9 and S10 Videos). Additionally, such features could not be detected in GFP-Rab11a DN$^{low}$ cells, corroborating that a functionally active Rab11a is key for viral inclusion formation (S2A Fig and S7, S8, S9 and S10 Videos).

By performing photomontages of single sections covering an entire plane of the cell using TEM, as demonstrated in Fig 2E, we manually scored the average number of single- and double-membrane vesicles per cell section (in a total of 10 distinct cells). We observed that both the number of single- (smv; Fig 2F) and double-membrane vesicles (dmv; Fig 2G) increased throughout infection, with statistically significant differences for the latter at 12 h and 16 h of infection (mean number of dmv ± SEM: Mock: 0.0 ± 0.0, 12 h– 5.1 ± 1.1, 16 h– 5.3 ± 1.6).

We also confirmed the presence of GFP-Rab11a and vRNPs in viral inclusions by Tokuyasu double immunogold labeling (S2B Fig) using antibodies against, respectively, GFP and the viral NP protein (which coats vRNPs). The single-membrane vesicles (smv, green arrowhead) stained positive for GFP-Rab11a (18 nm gold particle) and vRNPs (10 nm gold particle), whereas the double-membrane vesicles (dmv, yellow arrowhead) stained mostly for vRNPs. In mock-infected cells, no aggregation of single-membrane vesicles positive for Rab11a was observed, and vRNPs were not detected (S2B Fig).

We conclude that viral inclusions are biomolecular condensates positive for Rab11a and vRNPs, which contain inside double-membrane vesicles and numerous single-membrane vesicles of heterogeneous sizes that concentrate close to the ER. This type of condensate containing membranes inside is similar to condensates of phase separated synaptic vesicles, which form for ready deployment during neurotransmission release [25]. Within viral inclusions, single-membrane vesicles are likely Rab11a-positive endosomes, whereas double-membrane vesicles may be products of autophagy.

## ATG9A regulates the formation of viral inclusions

The presence of double-membrane vesicles suggests that IAV inclusion formation may depend on autophagy activation or manipulation of specific autophagy machinery, as has been proposed for other viruses that establish double-membrane vesicles for replication (reviewed in

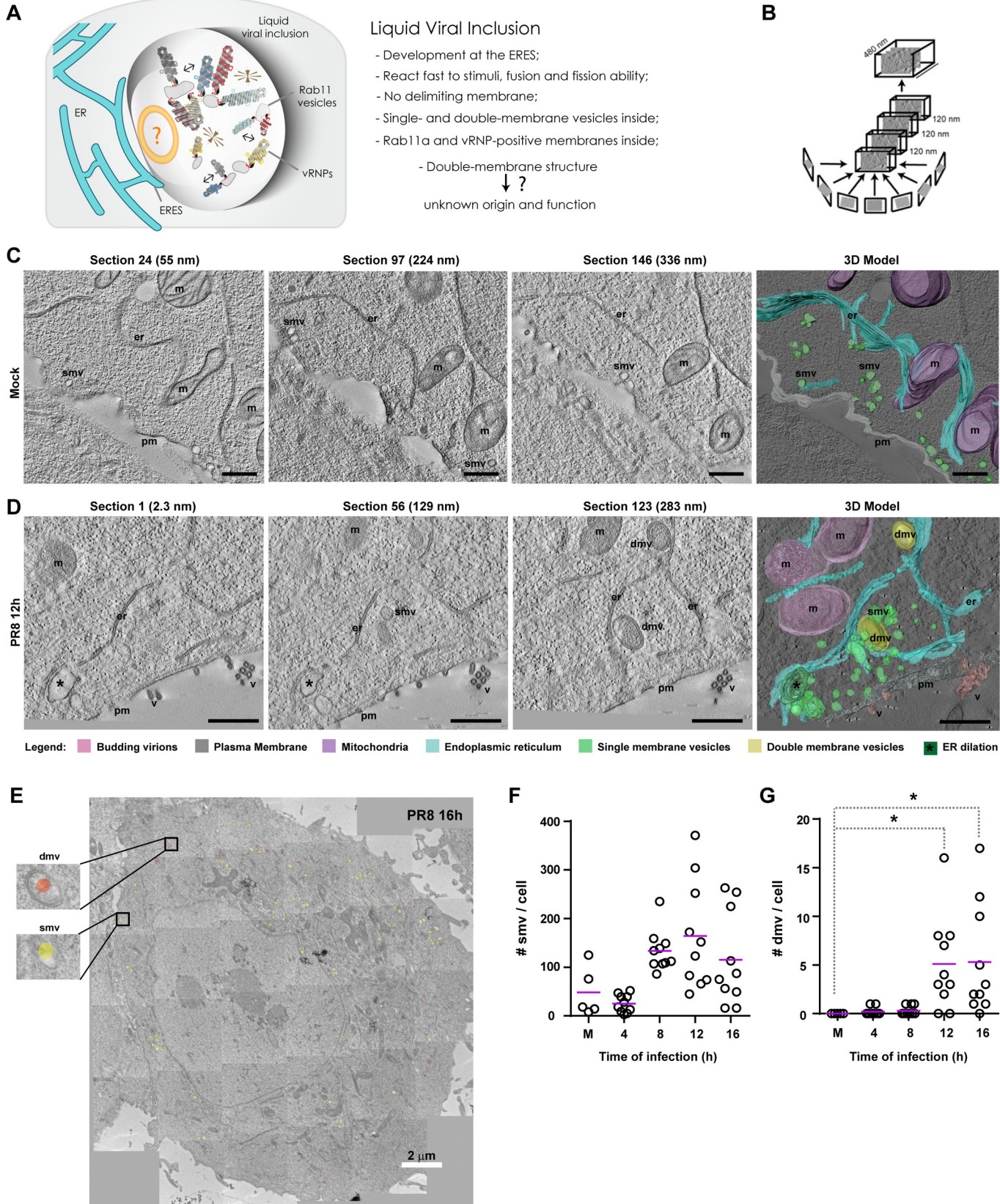

**Fig 2. Viral inclusions are biomolecular condensates containing single- and double-membranes inside. (A)** Schematic representation of a liquid viral inclusion according to 2D light and electron microscopy analysis. **(B)** Schematic representation of how 4 sequential tomograms (of 120 nm each) were acquired and stitched together (approximately 480 nm total thickness). **(C, D)** Cells (GFP-Rab11a WT$^{low}$) were infected or mock-infected with PR8 virus for 12 h at an MOI of 3. Cells were processed by high-pressure freezing/freeze substitution and imaged by ET-TEM. Representative cells are shown with 3 individual sections (including section height in nm) and the 3D cumulative model. Bar = 500 nm. Images were extracted from S3, S4, S5, and S6 Videos. Abbreviations: pm, plasma membrane (gray); er, endoplasmic reticulum (blue); v, budding virions (pink); m, mitochondria (purple); smv, single-membrane vesicle (light green); dmv, double-membrane vesicle (yellow); *, ER dilation (dark green). **(E)** Photomontages of single sections covering an entire plane of the cell were acquired by TEM at several times postinfection (4–16 h), as exemplified here at 16 h postinfection. The photomontages were used to score the number of single (smv) and double (dmv) membrane vesicles in **(F)** and **(G)**. Bar = 2 μm. **(F, G)** The number of single (smv) and double (dmv) membrane vesicles was manually scored and plotted as a function of time of infection. Statistical analysis was done by Kruskal–Wallis test (*$p < 0.05$). On average, 10 cells were analyzed per condition **(C-G)**. Experiments were performed twice. All the values of individual and pooled experiments are provided in S1 Data File. ER, endoplasmic reticulum; ET, electron tomography; MOI, multiplicity of infection; TEM, transmission electron microscopy; 2D, 2-dimensional; 3D, 3-dimensional.

[34]). Hence, we performed a small siRNA screening to test if key autophagy factors were necessary for viral replication and for viral inclusion development. The autophagy factors included canonical autophagy initiation proteins (ULK1 and ULK2, Unc-51 like autophagy activating kinase 1/2), membrane remodelers during autophagosome formation (ATG2A and ATG9A, autophagy related gene 2A and 9A), or negative regulators of membrane delivery from Rab11-recycling endosomes to forming autophagosomes (TBC1D14, Tre2/Bub2/Cdc16 1 domain-containing protein 14) [35–38].

For this screening, A549 cells were treated for 48 h with siRNA non-targeting (siNT) and siRNA targeting the above autophagy factors (siULK1, siULK2, siATG2A, siATG9A, siTBC1D14) and then infected or mock-infected with PR8 virus for 8 h (Fig 3). We tested how depletion of such factors impacted viral production by plaque assay, and viral inclusion formation by immunofluorescence using host Rab11a (gray) and viral NP protein (magenta) as viral inclusion markers.

Depletion of ULK1, TBC1D14, and ATG2A did not affect viral production (Fig 3A, mean PFU.mL$^{-1}$ ± SEM: siNT 1,686,000 ± 210,926, siULK1 1,600,000 ± 208,167, siTBC1D14 1,426,000 ± 12,333, siATG2A 1,567,000 ± 66,667) nor induced changes in viral inclusion shape (Figs 3D and S3B). For these factors, depletion caused a knockdown efficiency of 80% to 90% (Fig 3B and 3C) as measured by western blotting (Fig 3C, mean relative expression ± SEM: siNT 1.000 ± 0.000 versus siATG2A 0.083 ± 0.014) or quantitative reverse transcription PCR (RT-qPCR) (Fig 3B, mean relative expression ± SEM: siNT 1.000 ± 0.000, siULK1 0.248 ± 0.038, siTBC1D14 0.253 ± 0.015) for cases in which antibodies were not suitable. Although depletion of ULK2 led to a significant reduction in viral titers (1.5 log, Fig 3A, mean PFU.mL$^{-1}$ ± SEM: siNT 1,686,000 ± 210,926 versus siULK2 67,667 ± 3,930), it did not affect viral inclusion development (Figs 3D and S3B). This indicated that ULK2 may have a role in other steps of the viral lifecycle or that the siRNA used for ULK2 depletion caused an off-target effect. We are currently investigating the role of ULK2 in IAV infection but will not show that data here as it is not the focus of the present study. Moreover, ULK2 depletion was around 50% as determined by RT-qPCR (Fig 3B, mean relative expression ± SEM: siNT 1.000 ± 0.000 versus siULK2 0.526 ± 0.030), and we could not find a suitable antibody to specifically detect this protein by western blotting.

Depletion of ATG9A led to a 0.6 log drop in viral titers (Fig 3A, mean PFU.mL$^{-1}$ ± SEM: siNT 1,686,000 ± 210,926 versus siATG9A 408,333 ± 88,409) and led to a protein knockdown efficiency of approximately 80% as quantified by western blotting (Fig 3C, mean relative expression ± SEM: siNT 1.000 ± 0.000 versus siATG9A 0.197 ± 0.038). Numerous ATG9A bands can be observed on the blot, which are likely glycosylated or phosphorylated forms of this protein [39].

Alongside the drop in viral production, we also observed an alteration in the shape of viral inclusions. In control cells, vRNPs and Rab11a endosomes aggregated into rounded viral

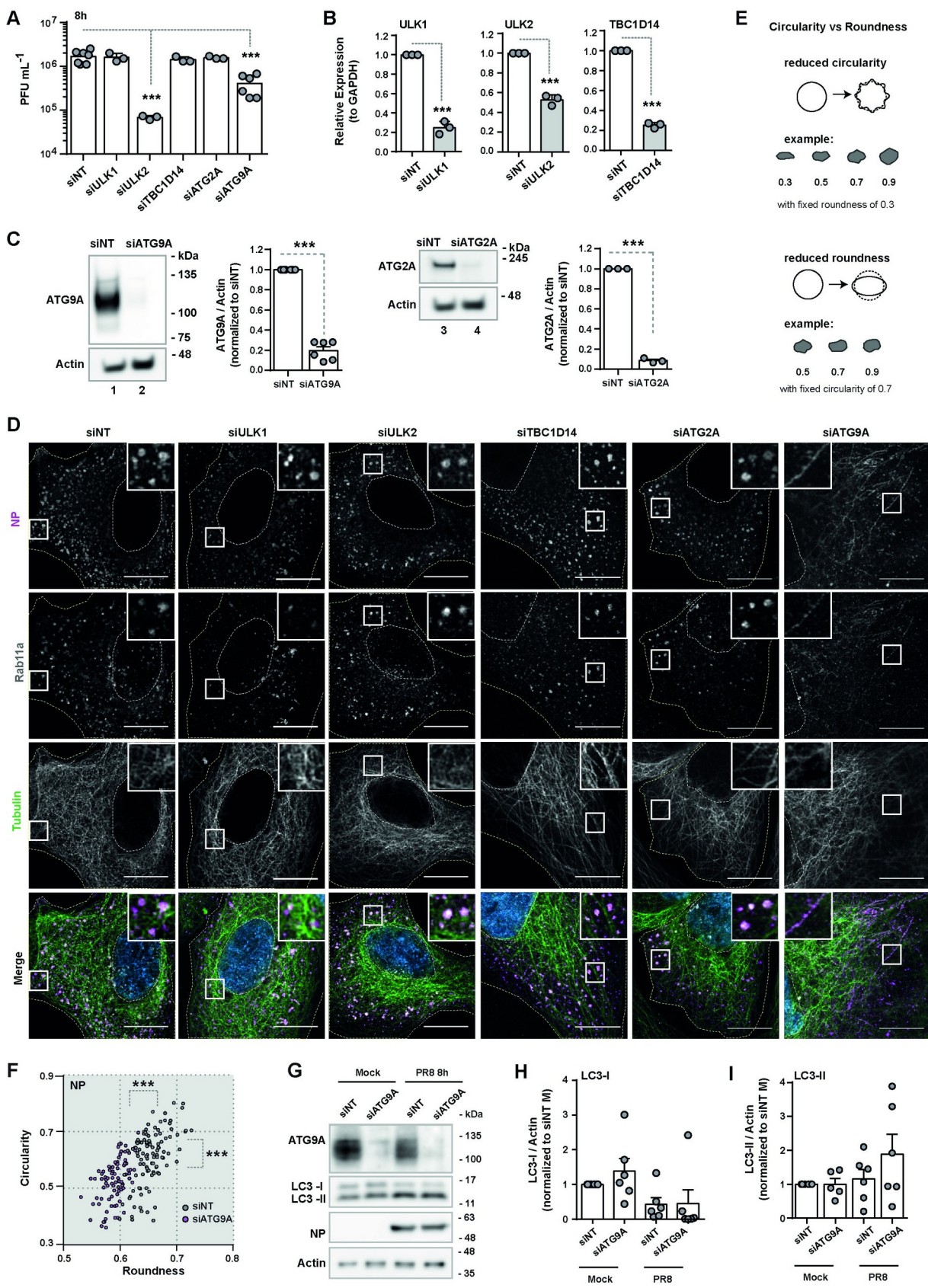

**Fig 3. ATG9A is determinant for the correct shape of IAV inclusions.** Cells (A549) were treated with siRNA non-targeting (siNT) or targeting ULK1 (siULK1), ULK2 (siULK2), TBC1D14 (siTBC1D14), ATG2A (siATG2A), or ATG9A (siATG9A) for 48 h and then infected or mock-infected (M) with PR8 virus for 8 h, at an MOI of 3. **(A)** Viral production was determined at 8 h postinfection by plaque assay and plotted as PFU per mL ± SEM. Data were pooled from 3–6 independent experiments. Statistical analysis was done by one-way ANOVA, followed by a Dunnett's multiple comparisons test (***$p < 0.001$). **(B)** The mRNA level of ULK1/2 and TBC1D14 before infection was quantified by real-time RT-qPCR and plotted as the relative expression to GAPDH mRNA level ± SEM. Expression was normalized to siNT from mock-infected cells. Data are a pool from 3 independent experiments. Statistical analysis was done by Student t test (***$p < 0.01$). **(C)** Protein levels of ATG2A and ATG9A before infection were determined by western blotting and plotted as the relative expression to actin protein levels ± SEM. Expression was normalized to siNT from mock-infected cells. Data are a pool from 3–6 independent experiments. Statistical analysis was done by Student t test (***$p < 0.01$). The original uncropped blots can be found in S1 Raw Images. **(D)** Localisation of Rab11a (gray), tubulin (green), and viral NP (magenta) proteins at 8 h postinfection was determined by immunofluorescence using antibody staining. Viral inclusions/vRNPs/Rab11a are highlighted by white boxes. Cell periphery and nuclei (blue, Hoechst staining) are delineated by yellow and white dashed lines, respectively. Bar = 10 μm. The figure panels for the corresponding mock-infected cells can be found in S3A Fig. **(E)** A schematic representation of shape classification based on circularity versus roundness is shown. **(F)** The roundness and circularity of viral inclusions/vRNPs, marked by NP staining, were determined at 8 h postinfection using the Shape Descriptor tool (Image J, NIH) and plotted against each other for siNT and siATG9A-treated cells. The maximum value of roundness and circularity (1) corresponds to a circular structure, whereas the minimum value represents a linear structure (0). More than 80 cells, pooled from 3 independent experiments, were analyzed per condition. Statistical analysis was done by Mann–Whitney test (***$p < 0.001$). A similar analysis done for the other autophagy factors is shown in S3B Fig. The frequency distribution of roundness and circularity of viral inclusions/vRNPs is shown in S4A and S4B Fig. **(G-I)** Protein levels of LC3-I and LC3-II were quantified by western blotting and plotted as the relative expression to actin protein levels ± SEM. Expression was normalized to siNT from mock-infected (M) cells. Data are a pool from 6 independent experiments. Statistical analysis was done by one-way ANOVA, followed by a Tukey's multiple comparisons test (no statistical significance detected). All the values of individual and pooled experiments are provided in S1 Data File, and the original uncropped blots can be found in S1 Raw Images. IAV, influenza A virus; MOI, multiplicity of infection; NP, nucleoprotein; PFU, plaque-forming unit; RT-qPCR, quantitative reverse transcription PCR; SEM, standard error of the mean; vRNP, viral ribonucleoprotein.

inclusions but formed instead a tubular network in cells depleted of ATG9A (Fig 3D). To express quantitatively the shape alterations (using ImageJ; Fig 3E), we plotted roundness versus circularity in both experimental conditions (Figs 3F and S3B). We used NP antibody staining to segment viral inclusions and quantify circularity/roundness, given that it produces a reduced signal-to-noise ratio compared to Rab11a antibody staining. The maximum value of circularity (1) corresponds to a perfect circle, whereas smaller values (approaching 0) correspond to shapes with a lower ratio of area to perimeter (long and irregular shapes or rough indented angular surfaces). Roundness (maximum value of 1 and minimum of 0) discriminates structures with circular cross-section from those with different geometric shapes (ellipses, rectangles, and irregular shapes). By plotting circularity versus roundness, we could better describe how the shape of viral inclusions changed upon depletion of ATG9A, as illustrated in the schematic representation (Fig 3E; adapted from [40]). The viral inclusions in siNT-treated cells had circularity values ranging from 0.39 to 0.80 with 95% confidence interval of [0.59 to 0.63], whereas in siATG9A-treated cells, the vRNP/Rab11a tubulated network values ranged from 0.36 to 0.70 with 95% confidence interval of [0.51 to 0.54] (Fig 3F). The viral inclusions in siNT-treated cells had roundness values ranging from 0.60 to 0.73 with 95% confidence interval of [0.65 to 0.66], whereas in siATG9A-treated cells, the vRNP/Rab11a tubulated network values ranged from 0.53 to 0.63 with 95% confidence interval of [0.59 to 0.60] (Fig 3F).

Calculation of the frequency distribution of circularity and roundness also clearly showed that viral inclusions in control cells were skewed toward a circular shape, whereas the structures in ATG9A-depleted cells were skewed toward a linear shape (S4A and S4B Fig). This result strongly supports our observation that ATG9A-depleted cells form tubular structures positive for vRNPs and Rab11a.

From this siRNA screening of host factors impacting IAV infection, ATG9A stood out as a putative candidate to explain viral inclusion formation (Fig 3). This lipid scramblase supplies membrane from donor organelles like the ER, Golgi, or the recycling endosome to the autophagosome [35–37,41–46]. Mechanistically, ATG9A flips phospholipids between 2 membrane leaflets, thus contributing to membrane growth [26]. Although ATG9A was initially identified

as a core member of the autophagic machinery, novel roles unrelated to autophagy have been discovered recently, including plasma membrane repair [27], lipid mobilization between organelles [28], and regulation of innate immunity [29]. To address whether the role of ATG9A in viral inclusion formation was related to autophagy pathway, we performed a western blotting to detect the levels of LC3 lipidation in cells depleted or not of ATG9A and subsequently infected or mock-infected with PR8 virus for 8 h (Fig 3G–3I). We could not observe any statistically significant differences in the levels of LC3-II in all tested conditions, suggesting that the effect of ATG9A in viral inclusions is unrelated to activation of the full autophagy pathway (mean relative expression LC3-II ± SEM: siNT Mock 1.000 ± 0.000, siATG9A Mock 0.996 ± 0.390, siNT PR8 1.162 ± 0.281, siATG9A PR8 1.885 ± 0.583).

In sum, we conclude that ATG9A regulates the formation of liquid IAV inclusions. In the absence of ATG9A, vRNPs and Rab11a do not aggregate into the characteristic rounded viral inclusions but instead form a tubular network scattered throughout the cell.

## ATG9A is mobilized from the Golgi during IAV infection

Although the major contribution for the expansion of the ERES membrane comes from the ER-Golgi vesicular cycling [35,37,41], whose impairment prevents IAV inclusion formation [4], recent evidence points toward the recycling endosome as an additional ATG9A reservoir and membrane donor compartment [42,43]. Given this, we sought to determine the donor compartment from which ATG9A is mobilized during IAV infection—the Golgi or the recycling endosome [43–45]. We confirmed that, in mock-infected cells, the major pool of ATG9A (green) colocalized with the Golgi matrix protein GM130 (gray), in agreement with published data [46], and no staining was detected in cells depleted of ATG9A (S5A Fig). However, we observed that ATG9A presented a cytoplasmic (possibly perinuclear) staining in PR8 virus–infected cells, which no longer colocalized with the Golgi (S5A Fig). Infection induced a gradual loss of ATG9A from the Golgi (Fig 4A), as the colocalization between ATG9A and Golgi matrix protein GM130 decreased throughout infection (Fig 4A and 4B, mean ± SEM of Pearson R value: Mock 0.411 ± 0.015, 4 h 0.400 ± 0.015, 6 h 0.318 ± 0.018, 8 h 0.281 ± 0.014, 14 h 0.242 ± 0.015). Moreover, we showed that the absence of ATG9A staining at the Golgi at later stages of infection is due to protein relocation and is not due to degradation. As can be appreciated from the western blot (Fig 4C), the total ATG9A protein levels remained constant throughout infection (mean ± SEM of Pearson R value: Mock 4 h 1.000 ± 0.000, 4 h 0.893 ± 0.494, Mock 6 h 0.693 ± 0.153, 6 h 1.387 ± 0.130, Mock 8 h 1.037 ± 0.194, 8 h 0.893 ± 0.039, Mock 14 h 1.160 ± 0.257, 14 h 1.060 ± 0.445).

We could not detect the subcellular location of endogenous ATG9A upon leaving the Golgi in infected cells. This could be due to the fact that ATG9A redistribution dilutes protein levels that are harder to detect using antibody staining. Alternatively, we detected the localization of ATG9A in overexpression experiments by transfecting A549 cells with a plasmid encoding GFP-ATG9A (Fig 4D, 4E and 4F) or GFP (as control; S5B and S5C Fig) and infecting them with PR8 virus for 8 h. Even though we acknowledge the limitations of overexpression experiments, in the sense that may change the cell and lead to an alteration of the cellular distribution of ATG9A, we could confirm that GFP-ATG9A strongly localized at the Golgi in mock-infected cells and that this colocalization is lost with infection (Fig 4D), as observed for endogenous ATG9 (Fig 4A). This observation reinforces the data on GFP-ATG9A establishing multiple contacts with viral inclusions identified by NP and Rab11a (Fig 4E). This pattern resembles the one we previously described using ERES markers (Sec16 and Sec31) [4]. Moreover, ATG9A puncta were found in the vicinity of viral inclusions and the ER (inset in Fig 4F).

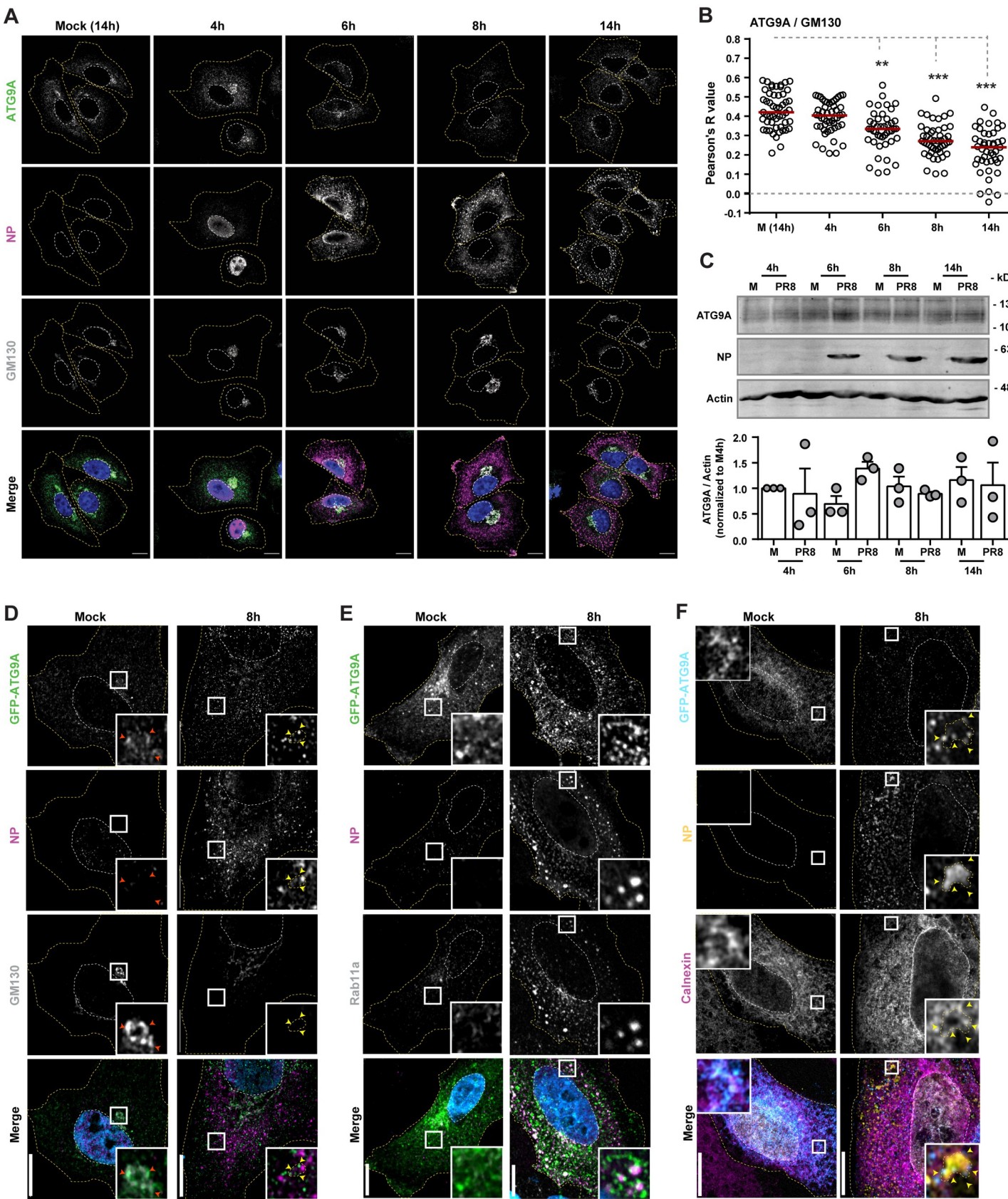

**Fig 4. ATG9A is mobilized from the Golgi/TGN during IAV infection. (A–C)** Cells (A549) were infected or mock-infected with PR8 virus, at an MOI of 3, for the indicated times. **(A)** The localization of host proteins ATG9A (green) and GM130 (gray) and viral protein NP (magenta) was determined by immunofluorescence using antibodies against these proteins. Mock-infected cells were collected at the same time as the 14 h-infected cells. Nuclei (blue, Hoechst staining) and cell periphery are delimited by white and yellow dashed lines, respectively. Bar = 10 μm. **(B)** Colocalization between ATG9A and GM130 in the images acquired in **(A)** was determined using the Colocalization Threshold analysis tool (FIJI/Image J, NIH) and plotted as the Pearson R value. Approximately 30 cells, from a single experiment, were analyzed per experimental condition. Red bar represents the median of values. Statistical analysis was done by Kruskal–Wallis test (**$p > 0.01$; ***$p > 0.001$). **(C)** The levels of ATG9A, actin, and viral NP protein in cell lysates at the indicated time points were determined by western blotting. ATG9A band intensity was quantified using FIJI (ImageJ, NIH) and normalized to actin levels. Original blots can be found in S1 Raw Images. Experiments **(A–C)** were performed twice. **(D–F)** Cells (A549) were transfected with a plasmid encoding GFP-ATG9A for 24 h and then infected or mock-infected with PR8 virus, at an MOI of 10, for 8 h. The localization of endogenous host proteins (GM130 –Golgi, Calnexin–ER, or Rab11a –recycling endosome) and viral protein NP was determined by immunofluorescence using antibodies against these proteins. Nuclei (blue or gray, Hoechst staining) and cell periphery are delimited by white and yellow dashed lines, respectively. Yellow arrowheads highlight areas of contact between viral inclusions and overexpressed GFP-ATG9A protein. Red arrowheads highlight areas of colocalization between GFP-ATG9A and the Golgi marker GM130. Bar = 10 μm. Control cells expressing GFP alone can be found in S5B and S5C Fig. Experiments **(D–F)** were performed twice. All the values of individual and pooled experiments are provided in S1 Data File. ER, endoplasmic reticulum; GFP, green fluorescent protein; IAV, influenza A virus; MOI, multiplicity of infection; NP, nucleoprotein; TGN, *trans*-Golgi network.

Cells overexpressing GFP alone were similarly infected, and the morphology or distribution of the ER and Golgi were also not significantly affected (S5B and S5C Fig).

We conclude that ATG9A is mobilized from the Golgi upon IAV infection and can be found surrounding viral inclusions close to the ER.

## ATG9A impacts viral inclusion formation without affecting the binding of vRNPs to the recycling endosome

Given that the recycling endosome could also be a putative source of ATG9A [42,43] during IAV infection and that both ATG9A and Rab11a could act in concert to allow the formation of viral inclusions, we tested the effect of depleting ATG9A in cells expressing a functionally active (WT) or inactive (DN) Rab11a. Cells expressing GFP-Rab11a WT$^{low}$ or GFP-Rab11a DN$^{low}$ were treated with siRNA non-targeting (siNT) or targeting ATG9A (siATG9A) for 48 h and then infected or mock-infected with PR8 virus for 10 h. In this case, we explored the link between Rab11a and ATG9A at 10 h after infection, as the GFP-Rab11 DN$^{low}$ cells produce low levels of viral particles before this period (by plaque assay), as we have shown before [4]. We observed that the drop in viral titers caused by ATG9A depletion was identical (approximately 0.6 log) in both cell lines, indicating that the effect of ATG9A in IAV infection is independent from Rab11a (Fig 5A, mean PFU.mL$^{-1}$ ± SEM: siNT Rab11a WT 908,333 ± 177,678, siATG9A Rab11a WT 195,000 ± 18,394, siNT Rab11a DN 1,612 ± 333, siATG9A Rab11a DN 320 ± 85). We also confirmed that the efficiency of ATG9A depletion was above 80% for both cell lines (Fig 5B, mean relative expression ± SEM: siNT Rab11a WT 1.000 ± 0.000; siATG9A Rab11a WT 0.1067 ± 0.027; siNT Rab11a DN 1.000 ± 0.000; siATG9A Rab11a DN 0.180 ± 0.090). As observed before [4], introducing GFP-Rab11a DN$^{low}$ exogenously in cells resulted in a 2.8 log difference (Fig 5A, mean PFU.mL$^{-1}$ ± SEM: siNT Rab11a WT 908,333 ± 177,678 versus siNT Rab11a DN 1,612 ± 333) in viral titers relative to the introduction of GFP-Rab11a WT$^{low}$. By immunofluorescence, we verified that Rab11a-positive structures were also tubular upon ATG9A depletion in GFP-Rab11a WT$^{low}$ cells (Fig 5C). On the contrary, GFP-Rab11a DN$^{low}$ cells do not establish viral inclusions (S1B Fig) as previously shown [4], regardless of the presence of ATG9A (Fig 5C).

We hypothesized that vRNP tubulation caused by ATG9A depletion could be due to the lack of vRNP association to Rab11a endosomes. To test this, the distribution of vRNPs and Rab11a vesicles was detected by immunofluorescence using antibodies against viral NP (magenta) and the host Rab11a (green), respectively. We observed that, although ATG9A depletion induced vRNP tubulation, it did not interfere with the association between vRNPs and Rab11a endosomes (Fig 5D), as NP and Rab11a colocalize in both siNT and siATG9A-

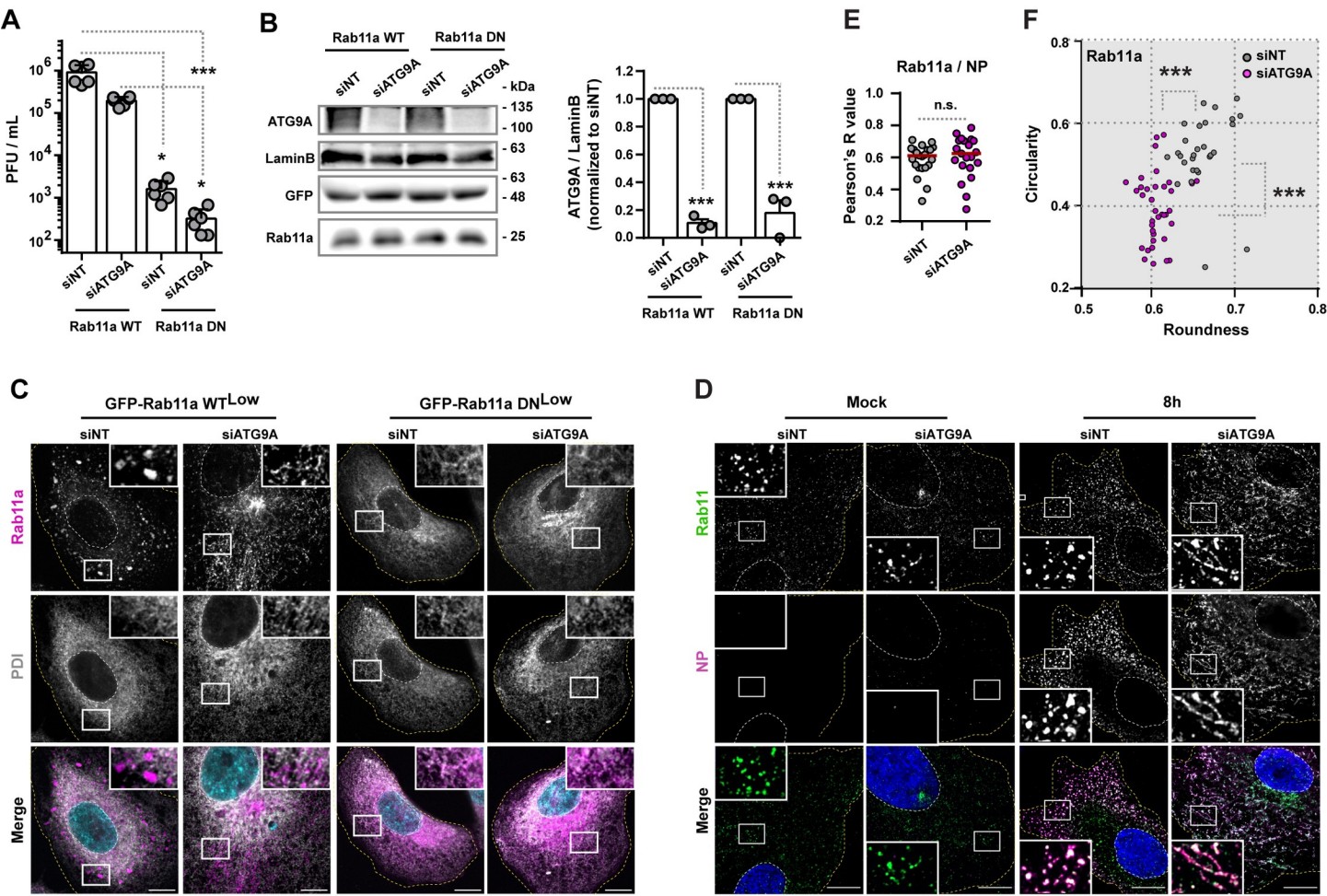

**Fig 5. ATG9A impacts viral inclusions independently of Rab11a–vRNP interaction. (A–C)** Cells (GFP-Rab11a WT$^{low}$ or GFP-Rab11a DN$^{low}$) were treated with siRNA non-targeting (siNT) or targeting ATG9A (siATG9A) for 48 h and then infected or mock-infected with PR8 virus for 10 h, at an MOI of 3. **(A)** Viral production was determined by plaque assay and plotted as PFU per milliliter (mL) ± SEM. Data represent 6 replicates from a single experiment. Two independent experiments were performed. Statistical analysis was done by one-way ANOVA, followed by a Kruskal–Wallis test (*$p < 0.05$; ***$p < 0.001$). **(B)** The protein level of ATG9A, lamin B, GFP, and Rab11a before infection were quantified by western blotting. The levels of ATG9A were plotted as the relative expression to lamin B level ± SEM. Expression was normalized to siNT from mock-infected cells. The data are a pool from 3 independent experiments. Statistical analysis was done by unpaired $t$ test between siNT vs. siATG9A conditions of each condition (Rab11a WT vs. DN mock; ***$p < 0.01$). **(C)** Localisation of Rab11a (magenta) and PDI (gray) at 10 h postinfection was determined by immunofluorescence using antibody staining. Viral inclusions/Rab11a are highlighted by white boxes. Cell periphery and nuclei (blue, Hoechst staining) are delineated by yellow and white dashed lines, respectively. Mock-infected cells can be found in S1B Fig. Bar = 10 μm. **(D–F)** Cells (A549) were treated with siRNA non-targeting (siNT) or targeting ATG9A (siATG9A) for 48 h and then infected or mock-infected with PR8 virus for 8 h, at an MOI of 3. **(D)** The localisation of host Rab11a (green) and viral NP (magenta) proteins at 8 h postinfection was determined by immunofluorescence using antibody staining. Viral inclusions/vRNPs are highlighted by white boxes. Cell periphery and nuclei (blue, Hoechst staining) are delineated by yellow and white dashed lines, respectively. Bar = 10 μm. Experiments were performed twice. **(E)** Colocalization between Rab11a and NP in the images acquired in **(D)** was determined using the Colocalization Threshold analysis tool (Image J, NIH) and plotted as the Pearson R value. At least 20 cells, pooled from 2 independent experiments, were analyzed per experimental condition. Red bar represents the median of values. Statistical analysis was done by Mann–Whitney test (n.s., not significant). **(F)** The roundness and circularity of Rab11a structures in the images acquired in **(D)** were determined using the Shape Descriptor tool (Image J, NIH) and plotted against each other. The maximum value of roundness and circularity (1) corresponds to a circular structure, whereas the minimum value represents a linear structure (0). Approximately 30 cells, from 2 independent experiments, were analyzed per condition. Statistical analysis was done by Mann–Whitney test (***$p < 0.001$). The frequency distribution of roundness and circularity of structures marked by Rab11a is shown in S4C and S4D Fig. All the values of individual and pooled experiments are provided in S1 Data File. GFP, green fluorescent protein; IAV, influenza A virus; MOI, multiplicity of infection; NP, nucleoprotein; PFU, plaque-forming unit; SEM, standard error of the mean; vRNP, viral ribonucleoprotein.

treated cells (Fig 5E, mean Pearson R value ± SEM of: siNT 0.5855 ± 0.02015 versus siATG9A 0.6015 ± 0.0287). The quantification of the circularity versus roundness of structures marked by Rab11a showed that ATG9A depletion also caused their tubulation (Fig 5F), thus matching

the previous quantification made using NP (Fig 3F). The viral inclusions in siNT-treated cells had circularity values ranging from 0.26 to 0.66 with 95% confidence interval of [0.50 to 0.57], whereas Rab11a structures in siATG9A-treated cells values ranged from 0.25 to 0.57 with 95% confidence interval of [0.37 to 0.43] (Fig 5F). The viral inclusions in siNT-treated cells had roundness values ranging from 0.62 to 0.72 with 95% confidence interval of [0.65 to 0.67], whereas Rab11a structures in siATG9A-treated cells values ranged from 0.57 to 0.66 with 95% confidence interval of [0.60 to 0.61] (Fig 5F). Calculation of the frequency distribution of circularity and roundness, using Rab11a as marker, also showed that viral inclusions in control cells were skewed toward a circular shape, whereas Rab11a structures in ATG9A-depleted cells were skewed toward a linear shape (S4C and S4D Fig).

We conclude that ATG9A is critical for proper establishment of IAV inclusions and that, in its absence, these fail to form. This defect, however, is unlikely to be related to the association of vRNPs to Rab11a vesicles as ATG9A depletion did not interfere with the colocalization and spatially synchronized dynamic movement of vRNPs-Rab11.

## ATG9A impacts the affinity of viral inclusions to microtubules

Our finding that ATG9A depletion induced morphological changes on viral inclusions from circular to tubular that colocalized with tubulin (Fig 3D) strongly hinted that viral inclusions were moving on microtubules. To test if ATG9A influenced the trafficking of vRNPs and Rab11a on microtubules, we performed live cell imaging of GFP-Rab11a WT$^{low}$ cells treated with siRNA non-targeting (siNT) or targeting ATG9A (siATG9) for 48 h and then infected or mock-infected with PR8 virus for 8 h. Rab11a was used as a proxy to track movement of viral inclusions (magenta), whereas Sir-Tubulin dye was added at the time of infection to visualize microtubules (green). In siNT-infected cells, we observed a dynamic but transient movement of Rab11a endosomes on microtubules (Fig 6A and linescan plots; S11 Video). In fact, most of Rab11a endosomes exhibited confined random movements, with occasional fast movements that were both processive and saltatory, as expected from previous reports [7,16]. Rab11a endosomes could be seen hopping on and off from the microtubule network (yellow arrows on highlighted inlets) to likely promote the dynamic fusion and fission movements required to form viral inclusions [4]. In siATG9A-infected cells, we observed that most Rab11a endosomes were moving on microtubules and few Rab11a endosomes detached and accumulated in the cytosol (Fig 6B and linescan plots, yellow arrows on highlighted inlets; S12 Video). The data indicate that the high affinity of Rab11a endosomes to microtubules in cells depleted of ATG9A confers the tubulated shape observed. In mock-infected cells, fast and short-lived movements of Rab11a endosomes could be traced, regardless of the presence of ATG9A in the cell, and no tubulation could be detected (Fig 6C and 6D and linescan plots; S13 and S14 Videos).

To confirm specific trafficking of viral inclusions on microtubules in cells depleted of ATG9A, we performed an experiment as described above and added nocodazole—to induce disassembly of microtubules—2 h before imaging live cells. First, we confirmed our previous results in which in siNT-treated and infected cells, viral inclusions became larger with little motility upon nocodazole treatment (Fig 6E–6H) [5,7]. Second, in ATG9A-depleted and infected cells treated with nocodazole, tubulated viral inclusions also became rounded structures without significant motility (Fig 6E), suggesting that ATG9A depletion caused an arrest of viral inclusions at microtubules, as observed for siNT-treated cells. Given that Rab11a endosomes are transported on microtubules for normal functions in noninfected cells, we also observed an accumulation of Rab11a in the cytosol of mock-infected cells, regardless of the presence of ATG9A (Fig 6F). Moreover, immunofluorescence data indicate that depletion of

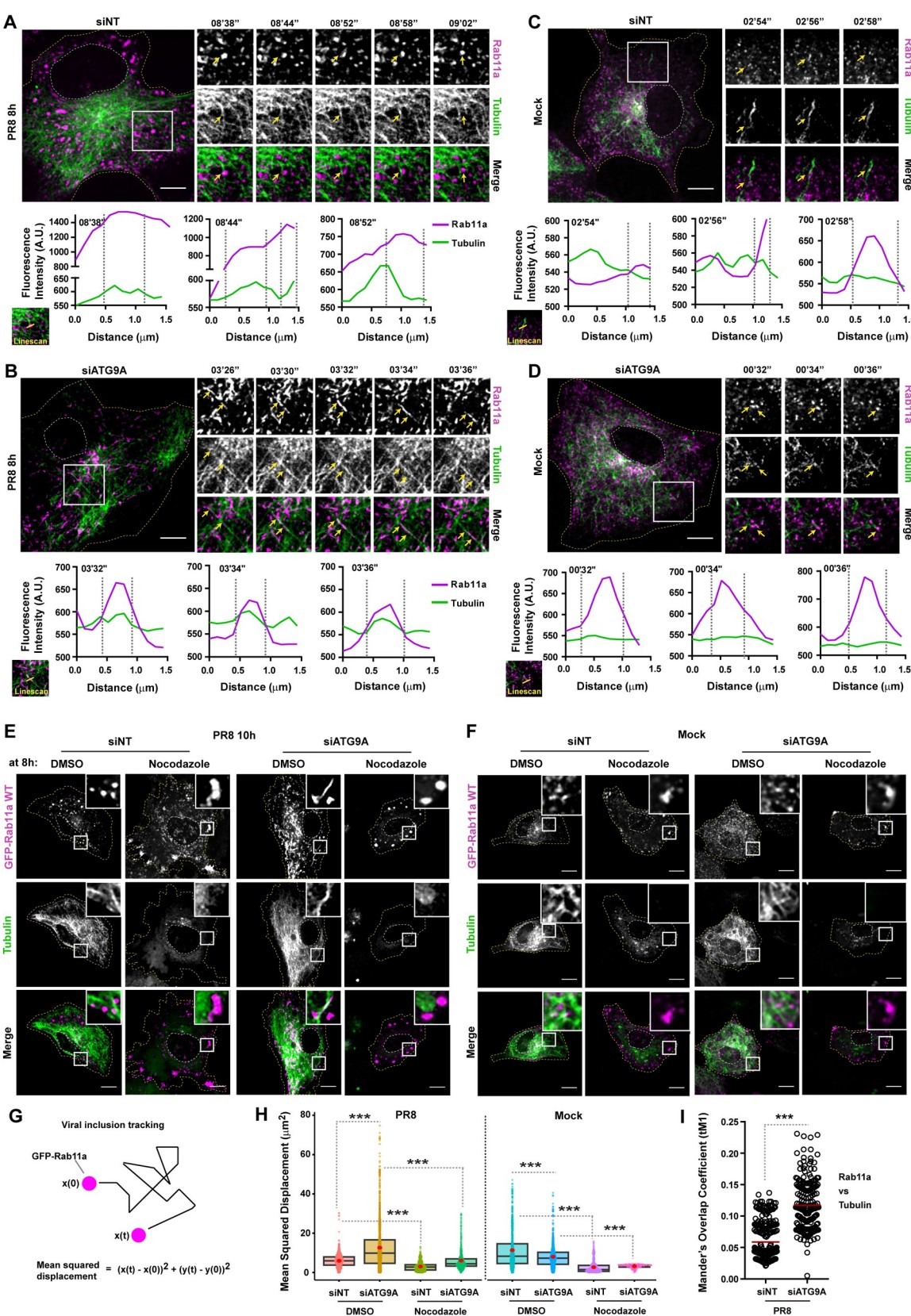

**Fig 6. ATG9A depletion arrests viral inclusions on microtubules. (A–D)** Cells (GFP-Rab11a WT$^{low}$, magenta) were treated with siRNA non-targeting (siNT) or targeting ATG9A (siATG9A) for 48 h. Upon this period, cells were infected or mock-infected with PR8 virus for 8 h, at an MOI of 3, and simultaneously treated with 200 nM Sir-Tubulin dye to stain the microtubules (green) in live cells. Cells were imaged for 10 min (2 s/frame) under time-lapse conditions at 8 h postinfection. White boxes show viral inclusions/Rab11a. Individual frames with single moving particles highlighted with yellow arrows are shown in the small panels. Bar = 10 μm. Images from selected infected cells were extracted from S11 and S12 Videos. Images from mock-infected cells were extracted from S13 and S14 Videos. For each case, a linescan was drawn as indicated to assess the dynamics of Rab11a and tubulin. The fluorescence intensity of Rab11a endosomes or viral inclusions (magenta) and tubulin (green) at indicated times was plotted against the distance (in μm). Representative analysis was performed using images from (**A-D**). (**E, F**) Cells (GFP-Rab11a WT$^{low}$) were treated as explained above (in A–D). At 8 h postinfection, cells were treated with DMSO or 10 μg/mL of nocodazole for 2 h. Cells were imaged at 10 h postinfection. White boxes show viral inclusions/Rab11a. Bar = 10 μm. (**G**) Scheme illustrates how viral inclusion/Rab11a endosome deviation from a reference position (in X and Y direction) was tracked by live cell imaging. The formula used to quantify the mean squared displacement (MSD, μm$^2$) is also shown. (**H**) Each viral inclusion/Rab11a endosome in a cell was tracked using the TrackMate plugin (FIJI, NIH) and displacement was quantified as explained in (**G**). Data were plotted as the MSD (μm$^2$) per treatment. The red dot indicates the median in the boxplots. Statistical analysis was done by a Kruskal–Wallis test (***$p < 0.001$). (**I**) Colocalization between microtubules (tubulin) and viral inclusions (Rab11a) in live cells was determined as the Manders' Overlap Coefficient tM1 (thresholded; explained in Methods section). Only infected conditions are shown. Given that very small Rab11a endosomes were scattered throughout the cytosol in mock-infected controls, we could not obtain reliable correlation coefficients. This result is, however, corroborated by quantifying colocalization between microtubules (tubulin) and viral inclusions using a fluorescent virus (PA-mNeonGreen PR8), as shown in S7 Fig. Between 6 and 10 cells per condition were analyzed. Statistical analysis was done by a Student $t$ test (***$p < 0.001$). Experiments were performed twice. All the values of individual and pooled experiments are provided in S1 Data File.

ATG9A did not affect the architecture of the microtubule network in either mock-infected or infected cells (Figs 6A, 6B, 6C, 6D and S6A).

Quantification of the mean squared displacement (MSD) of viral inclusions marked by Rab11a (Fig 6G and 6H) showed that their position significantly deviated) more with respect to the reference position in the absence of ATG9A than in the control (mean MSD ± SEM of: siNT 10 h 5.959 ± 0.003, siATG9A 10 h 12.626 ± 0.007, siNT Mock 11.331 ± 0.012, siATG9A Mock 7.998 ± 0.006). The MSD of viral inclusions upon nocodazole treatment was significantly impaired, regardless of the presence of ATG9A in the cell (mean MSD nocodazole ± SEM of: siNT 10 h 3.048 ± 0.003, siATG9A 10 h 6.047 ± 0.012, siNT Mock 2.677 ± 0.004, siATG9A Mock 3.152 ± 0.006). This result can be interpreted as Rab11a-vRNPs endosomes traveling longer distances in the absence of ATG9A and detaching less from microtubules, thus forming less rounded viral inclusions near the ER.

Moreover, we observed that depletion of ATG9A led to a higher colocalization between microtubules and viral inclusions in infected cells (Fig 6I; mean Manders' Overlap Coefficient ± SEM of: siNT 0.059 ± 0.029 versus siATG9A 0.117 ± 0.039). In mock-infected cells, small Rab11a endosomes are spread throughout the cytosol and, thus, strongly overlap with microtubules. For this reason, we obtained highly variable and unreproducible colocalization quantifications between experiments and, therefore, opted by not showing their values.

We confirmed that ATG9A specifically influenced the movement of viral inclusions on the microtubule network and not on the actin cytoskeleton (S6 Fig). To assess that, cells were treated with siNT or siATG9A and then were infected or mock-infected with PR8 virus for 8 h. By immunofluorescence, we stained for actin and microtubules in fixed cells using phalloidin or an antibody against tubulin, respectively. Similarly to our live cell imaging findings, vRNPs also colocalized with tubulin and presented a tubular shape in fixed cells depleted of ATG9A (S6A Fig). We could not detect significant colocalization of vRNPs on the actin cytoskeleton in either control or ATG9A-depleted cells. Also, ATG9A depletion did not impact the actin cytoskeleton architecture (S6B Fig).

Importantly, to corroborate that viral inclusions have higher affinity to microtubules in the absence of ATG9A, we depleted ATG9A and infected A549 cells with a fluorescent PR8 virus that expresses PA protein (a subunit of vRNPs) fused to mNeonGreen protein. This virus leads to productive infections and allows us to track vRNPs and, hence, viral inclusion formation by

live cell imaging [16]. At 8 h of infection, viral inclusions produced by PA-mNeonGreen were rounded in control cells and became tubular upon ATG9A depletion (S7A and S7B Fig, S15 and S16 Videos). When treated with nocodazole, viral inclusions in both experimental conditions became large, accumulated near the plasma membrane, and presented reduced motility, as expected (S7C and S7D Fig, S17 and S18 Videos). As before, we calculated the MSD of viral inclusions and their colocalization to microtubules (S7E and S7F Fig). We observed again that viral inclusions have increased displacement (mean MSD ± SEM of: siNT 10 h 2.144 ± 0.002, siATG9A 10 h 4.756 ± 0.005, siNT 10 h nocodazole 1.730 ± 0.005, siATG9A 10 h nocodazole 0.809 ± 0.001) and higher colocalization with microtubules (mean Manders' Overlap Coefficient ± SEM of: siNT 0.090 ± 0.065 versus siATG9A 0.150 ± 0.060) in the absence of ATG9A (S7E and S7F Fig), as seen with viral inclusions produced in GFP-Rab11a WT$^{low}$ cells infected with PR8 virus (Fig 6H).

Overall, our findings suggest that ATG9A influences the affinity of viral inclusions to the microtubule network. Although we could only detect the location of overexpressed ATG9A during IAV infection, we speculate that ATG9A might promote the transitioning of viral inclusions between microtubules and the ER.

## ATG9A impacts efficiency of viral genome assembly but not genome packaging into virions

Our previous findings showed that ATG9A influences the affinity of viral inclusions to microtubules. Given that viral inclusions are seen as the putative sites where the complex of the IAV segmented genome, comprising 8 different vRNPs, is formed (viral genome assembly), we hypothesized that the arrest of viral inclusions at microtubules caused by ATG9A depletion would affect late steps of viral infection. The late stages include genome assembly, viral surface protein levels, and the inclusion of the assembled genome (genome packaging) into budding virions. To test this hypothesis, we first quantified the number of vRNA copies of each viral segment by RT-qPCR in cells treated with siNT or siATG9A for 48 h and infected or mock-infected with PR8 virus for 8 h (Fig 7A). We observed that vRNA levels for the 8 segments were increased in cells depleted of ATG9A (2- to 3-fold increase), indicating that there was an accumulation of vRNPs inside the cell. This result was corroborated by quantifying the intensity/integrated density of viral NP protein, which coats the vRNA in vRNP particles, in the cell cytosol in both conditions upon immunofluorescence staining with an antibody against NP (Fig 7B).

We also quantified the levels of the 3 viral surface proteins—hemagglutinin (HA), neuraminidase (NA), and matrix protein 2 (M2)—at the plasma membrane of cells treated with siNT or siATG9A for 48 h and infected or mock-infected with PR8 virus for 8 h, using flow cytometry. We hypothesize that if there is accumulation of vRNPs in the cell cytosol that do not condense into viral inclusions near the ER, then less complete 8-vRNP genomes would be formed and would reach the surface budding sites. As a consequence, less virions would be released, and, as such, the surface viral proteins (HA, NA, and M2) would accumulate at the plasma membrane. Indeed, we observed that depletion of ATG9A led to a significant increase in the levels of the 3 viral proteins (35.9% for HA, 30.3% for NA, 40.2% for M2) at the plasma membrane (Fig 7C, median fluorescence intensity ± SEM of siNT versus siATG9A: HA– 8,674 ± 215 versus 13,534 ± 965; NA– 5,201 ± 114 versus 7466 ± 28; M2–3,064 ± 15 versus 5,127 ± 386).

In addition, we also tested whether the lack of ATG9A led to formation of virions containing an incorrect set of 8 vRNPs (both in number and in type) that, hence, would not be infectious. For this, we purified RNA from virions released from PR8 siNT- or siATG9A-infected

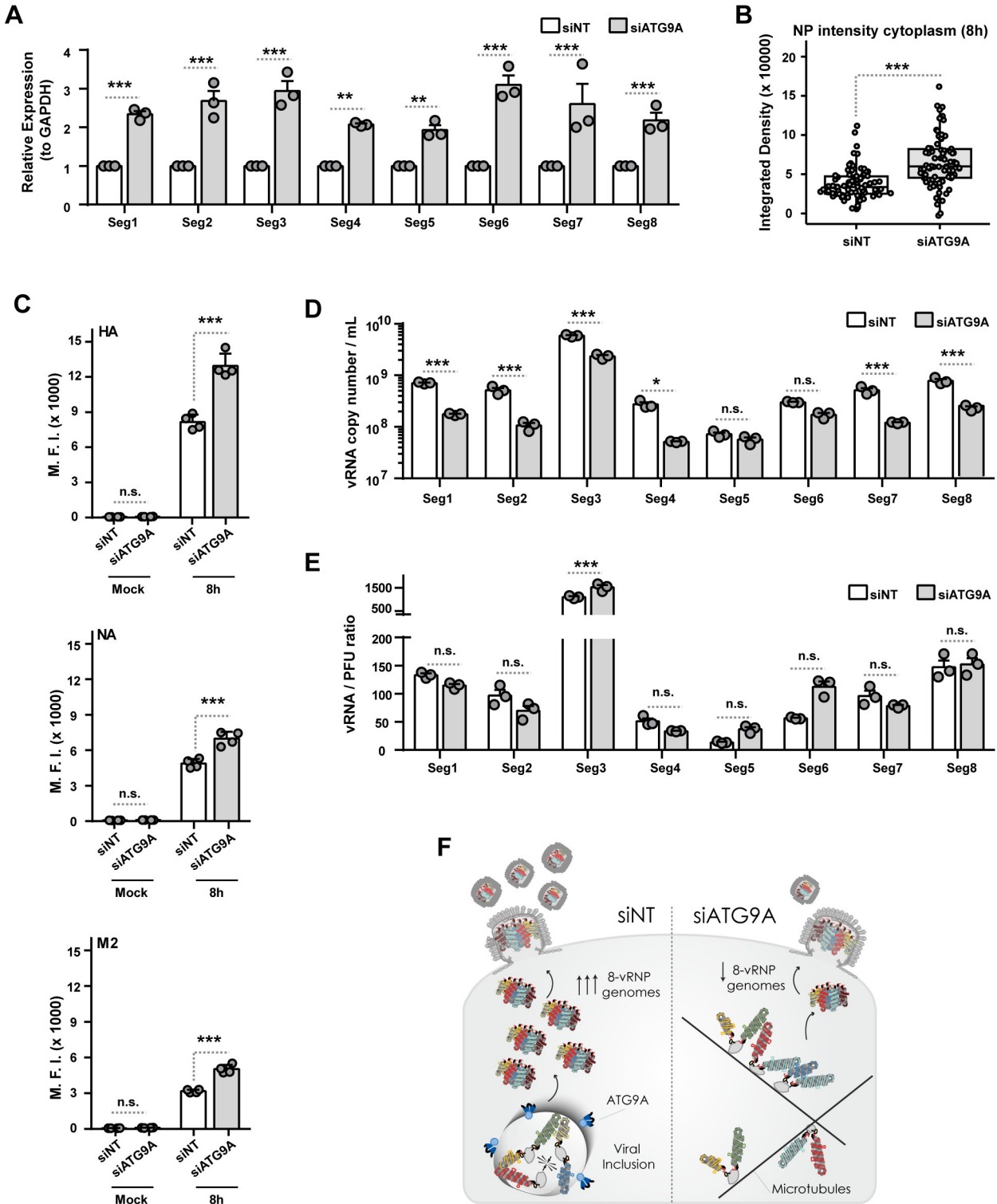

**Fig 7. ATG9A depletion affects viral genome assembly but not viral genome packaging.** Cells (A549) were treated with siRNA non-targeting (siNT) or targeting ATG9A (siATG9A) for 48 h and then infected or mock-infected with PR8 virus at an MOI of 3. **(A)** The vRNA levels for each viral RNA segment (1–8) was expressed relative to GAPDH levels at 8 h postinfection and was determined by real-time RT-qPCR using specific primers as detailed in the Methods section. Data are triplicates from a single experiment. Two independent experiments were performed. Statistical analysis was done by a two-way ANOVA test, followed by a Sidak's multiple comparisons test (**$p < 0.01$; ***$p < 0.001$). **(B)** The integrated density of NP protein in the cell cytosol at 8 h postinfection was determined by immunofluorescence, using the Analyze Particles function of FIJI

(ImageJ, NIH). More than 80 cells, pooled from 4 independent experiments, were analyzed per condition. Statistical analysis was done by Kruskal–Wallis test (***$p < 0.001$). **(C)** The levels of the 3 viral surface proteins at 8 h postinfection (HA, hemagglutinin; NA, neuraminidase; M2, matrix protein 2) were determined by flow cytometry using monoclonal antibodies against each viral protein and analyzed as shown. The MFI of each viral protein at the cell surface was plotted for each experimental condition. Statistical analysis was done by one-way ANOVA, followed by a Sidak's multiple comparisons test (**$p < 0.01$; ***$p < 0.001$). Data are a pool of 4 independent experiments performed. **(D, E)** The vRNA copy number per mL for each viral RNA segment (1–8) and the vRNA-to-PFU ratio at 8 h postinfection was determined by real-time RT-qPCR using specific primers as detailed in the Methods section. Data are triplicates from a single experiment. Two independent experiments were performed. Statistical analysis was done by a two-way ANOVA test, followed by a Bonferroni's multiple comparisons test (*$p < 0.05$; ***$p < 0.001$; n.s., not statistically significant). **(F)** Scheme illustrates that ATG9A is critical for viral inclusion shape and regulates rate of viral genome assembly but does not affect genome packaging into budding virions. All the values of individual and pooled experiments are provided in S1 Data File. MFI, median fluorescence intensity; MOI, multiplicity of infection; NP, nucleoprotein; PFU, plaque-forming unit; RT-qPCR, quantitative reverse transcription PCR.

cells for 8 h into the supernatant. Then, we quantified the number of vRNA copies as well as the vRNA-to-PFU ratio of each viral segment in both conditions, using RT-qPCR. If we observed a problem in genome packaging, although the levels of RNA would be similar in both conditions, an increase in vRNA-to-PFU ratio would be expected, as reported in [47]. Most vRNA segments had decreased copy numbers in virions from cells depleted of ATG9A, with the exception of segments 5 and 6 (Fig 7D and 7E), and the vRNA-to-PFU ratio did not significantly differ between the 2 conditions (Fig 7D and 7E). Overall, both results indicate that there was a decrease in the formation of complete 8-vRNP genomes, but not a major defect in their incorporation in virions in the absence of ATG9A (Fig 7F).

Taken together, these data suggest that ATG9A is likely involved in the regulation of viral inclusion distribution, facilitating circulation between microtubules and the ER. By interfering with viral inclusion trafficking, viral genome assembly efficiency is decreased and complete 8-vRNP genomes delivery to budding sites at the plasma membrane is also reduced, with concomitant accumulation of HA, NA, and M2 at the surface that is not incorporated as efficiently in budding virions. ATG9A may thus be a host catalyst that facilitates viral genome assembly (Fig 7F).

## Discussion

The importance of phase transitions to viral lifecycles has become evident in recent years, and knowledge on these processes may foster the design of innovative antivirals [48]. Many viruses that threaten public health (measles virus, herpes simplex virus 1, mumps virus, severe acute respiratory syndrome coronavirus 2, IAV, or human immunodeficiency virus) are able to establish functional biomolecular condensates to fulfill critical steps in their lifecycles, such as genome transcription, replication, virion assembly, and immune evasion [48]. In the case of IAV infection, viral inclusions with liquid properties arise close the ERES [4,5], behaving similarly to condensates described to form by liquid–liquid phase separation [19,48]. IAV liquid inclusions are viewed as key sites dedicated to viral genome assembly. Here, Rab11a and vRNPs concentrate and facilitate viral intersegment interactions [4–6].

Our present work contribute toward the current understanding of IAV genome assembly by uncovering a host factor, ATG9A, that mediates the exchange of viral inclusions between microtubules and the ER (Fig 8). In ATG9A-depleted cells, the change in shape and location of viral inclusions causes accumulation of vRNPs in the cytosol with concomitant reduction in the formation of 8-partite viral genomes and release of infectious virions. ATG9A contributes to the spatial distribution of viral inclusions and, thus, the ability of their main components, Rab11a and vRNPs, to demix from the cytosol (presumably by percolation associated with phase separation) at ER membranes. Whether the subcellular targeting of vRNPs using cellular machinery and the cytoskeleton allows vRNPs to reach the saturation concentration enabling phase separation remains unknown [4].

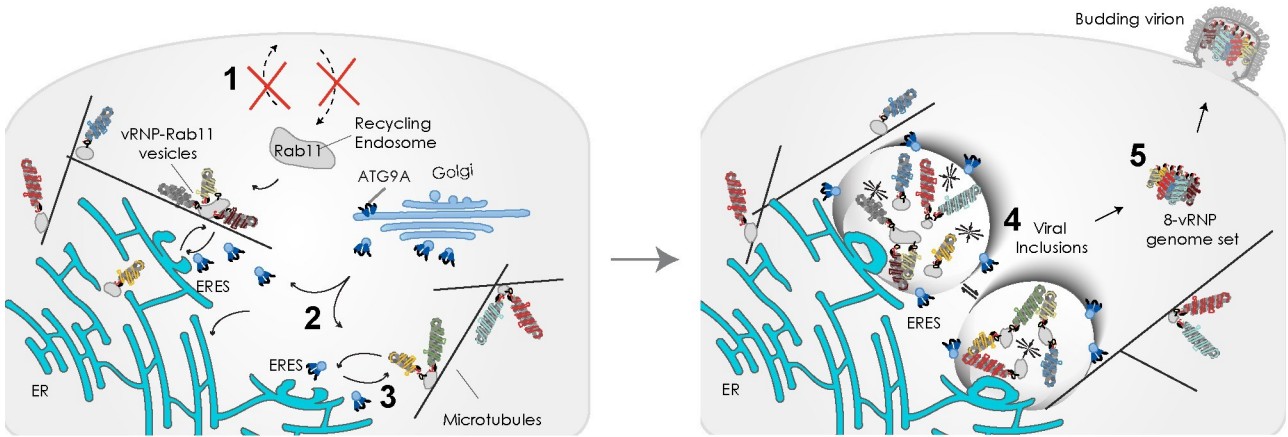

**Fig 8. Proposed model for ATG9A role in the establishment of liquid IAV inclusions.** We currently view liquid viral inclusions, composed of Rab11a endosomes and vRNPs, as sites dedicated to the assembly of the IAV genome [4,6,49]. We have previously shown that liquid viral inclusions develop in close contact with the ERES [4]. Here, we describe the initial events on the left panel that may lead to the formation of liquid viral inclusions on the right to facilitate the formation of IAV genomic complex. In this study, we demonstrate that IAV infection reduces the Rab11a-regulated recycling capacity of the host cell (step 1). This effect is likely a consequence of vRNP binding to Rab11a endosomes, which are then rerouted to the ERES to form viral inclusions. Such trafficking of Rab11a endosomes carrying the vRNPs to the ER is regulated by the host factor ATG9A. We identified that ATG9A is mobilized from the Golgi during IAV infection (step 2) and leads to the removal of Rab11a-vRNP complexes from microtubules when at the ER (step 3). It is thus possible that ATG9A moves to the ER to promote the linkage of viral inclusions to microtubules. In this location, vRNPs-Rab11a units may establish multiple and dynamic contacts forming liquid percolation-driven condensates. We also show (although with overexpression experiments) that ATG9A engages in multiple contacts with viral inclusions (step 4). We propose that the liquid properties of viral inclusions favor the formation of the 8-segmented IAV genome that is transported to the plasma membrane (step 5). ATG9A, autophagy related gene 9A; ER, endoplasmic reticulum; ERES, ER exit site; IAV, influenza A virus; vRNP, viral ribonucleoprotein.

Relevant to this field, in general, is to understand how exactly the transport of components regulates formation and activity of biomolecular condensates. Our proposed model is that during infection, progeny vRNPs attach outwardly to Rab11a recycling endosomes and are trafficked together, not to the surface as previously thought [7–10,13–15], but instead toward the ERES [4]. As infection progresses, vRNPs and Rab11a endosomes concentrate further at the ERES to form viral inclusions that share properties with bona fide liquid condensates [4,5]. A similar case is the clustering of synaptic vesicles in neurons that are organized by phase separation for ready deployment upon synaptic stimuli [25].

Why and how membrane-bound organelles such as the ER support condensate formation is being widely explored. It is well described that the ER forms contacts with many other organelles to modulate their biogenesis and dynamics, including liquid phase separated organelles (TIS granules, Sec and P-bodies, omegasomes) [19,21,50]. The ER may support many steps of the viral lifecycle. In fact, other authors have proposed that a remodeled ER membrane transports progeny vRNPs to the plasma membrane for viral packaging [30]. Alterations in ER shape in IAV infection could alternatively be linked to exploitation of lipid metabolism, deregulation of cell-autonomous immunity, or ER stress pathways [51]. Here, we found that viral inclusions displayed transient and highly dynamic movements at the ER (Fig 1D and 1E). These movements included fusion, fission, and sliding on the surface of the ER, like those we described for vRNPs [4]. In agreement, a recent study showed that the ER regulates the fission of liquid ribonucleoprotein granules for size control [21]. We presume that the ER could create a favorable environment where cellular machinery would support phase separation of viral

inclusions (Fig 1D), thus enabling efficient spatiotemporal coordination of IAV genome assembly.

Additionally, we also detected the presence of double-membrane vesicles in viral inclusions (Fig 2A–2G), which generally hints toward activation of autophagy in mammalian cells. With the exception of ATG9A, depletion of autophagy initiators (ULK1/2) or the partner of ATG9A (ATG2A) in membrane remodeling during autophagosome formation did not interfere with viral inclusion formation. Hence, we concluded that IAV is using specific autophagy machinery, ATG9A, for the assembly of viral inclusions (Fig 3D). Furthermore, ATG9A role in viral inclusion formation seems to be independent from full autophagy pathway activation, even though we acknowledge that the pathway may be initiated in infection. In fact, LC3 lipidation was shown during IAV infection [52], and, interestingly, it is described that in IAV infection, LC3 binds to a LIR motif on the viral M2 protein and is relocated to the plasma membrane, rather than concentrating on autophagosomes, and, hence, no full autophagy happens [52]. In our study, we found no differences in LC3 lipidation upon ATG9A depletion (Fig 3G–3I), suggesting that ATG9A depletion is not altering the pathway.

This work highlights that ATG9A is a versatile factor able to organize vesicular trafficking in ways that drive formation and activity of condensates by limiting access to microtubules. This finding allocates a new function to ATG9A beyond its well-known involvement in autophagy [44–46,53,54]. Recent studies support that ATG9A may be a key regulator of vesicular trafficking. ATG9A functions in protein export from the TGN [55], regulation of neurite outgrowth [56], in coupling autophagosome biogenesis to synaptic vesicle cycling [57], and chemotactic migration [58]. It is also involved in plasma membrane repair [27], lipid mobilization between organelles [28], as well as in the regulation of innate immunity [29]. These studies also show that ATG9A has a wide subcellular distribution according to the specific function being executed.

We found that ATG9A is mobilized from the Golgi/TGN during IAV infection (Fig 4A–4C) and establishes highly dynamic contacts with viral inclusions and the ER (Fig 4D–4F). This interaction was only detectable upon overexpression of ATG9A (Fig 4D–4F), which is a limitation of our study. We observed though that depleting ATG9A caused an arrest of Rab11a endosomes carrying vRNPs at microtubules (Fig 6A–6D). The dissociation of viral inclusions from microtubules is a regulated process, as vRNPs bound to Rab11a can be placed in microtubules when ATG9A is absent. This strongly suggests that the competition between vRNPs and molecular motors for binding to Rab11a, as initially proposed by us and others [6,9,16,18,59] may be a part of the process but is unable to explain the unbinding of Rab11a vesicles from microtubules fully. In this sense, ATG9A could catalyze the passage of viral inclusions to alternative transport means or locations such as the ER [4,30].

Given the highly dynamic nature of ATG9A and its ability to supply proteins and lipids from the Golgi (for example, phosphoinositide-metabolizing enzymes) [46,60], ATG9A may create ER microdomains favorable for phase separation of viral inclusions. This is in line with our previous finding that blocking the ER-Golgi vesicular cycling abolished the formation of liquid viral inclusions [4]. Although it was proposed that the recycling endosome is the primary reservoir of ATG9A for autophagosome initiation [42], we found that the main pool of ATG9A mobilized during IAV infection originated from the Golgi/TGN (Figs 2C and 3A) and that the effect of ATG9A was independent yet synergic with that of Rab11a in IAV infection. In fact, when overexpressed, ATG9A localizes between the ER and viral inclusions. This raises the hypothesis that when ATG9A is not functioning at the ER, vRNP-Rab11a endosomes circulate in microtubules but are unable to concentrate at ERES and, hence, do not form optimal viral inclusions. We think this effect is specific to IAV infection. Supporting the specificity of the observed effect is a recent study showing that IAV manipulates Rab11a endosome

transport by reducing its association to dynein [16]. We cannot exclude, however, the possibility that ATG9A interacts with other regulators of the recycling endosome, such as Rab11b.

In our study, we also addressed ATG9A significance and changes in liquid condensate shape to IAV production. We showed that, in the absence of ATG9A, there are less virions budding the cell, but released virions contain correct amounts of complete genomes (Fig 7D and 7E). This suggests that the cellular environment to create complete genomes is compromised when ATG9A is absent (viral inclusions fail to mount near the ER) and less complete genomes are produced (Fig 7F). In fact, we showed that there is a significant accumulation of all vRNP types in the cell cytosol upon ATG9A depletion (Fig 7A and 7B). This is consistent with a blockade in viral genome assembly distinct from viral transmembrane protein production and transport and points toward a defect in the efficiency of complete genomes formation in the cell. The accumulation of transmembrane proteins at the surface results from reduced budding and release of virions (Fig 7C).

The concept that ATG9A is a modulator of liquid–liquid phase separation on or near ER in mammalian cells has also been recently proposed by another group [22]. ATG9A regulates phase separation and spatial organization of the autophagosome component FIP200 on the ER tubules. Interestingly, FIP200 condensates associate and move along the ER strand and enlarge via growth or fusion, with ATG9A dynamically orbiting FIP200 condensates in a manner similar to viral inclusions (Fig 4D–4F). Whether ATG9A acts directly to locally coordinate phase separation of viral inclusions or indirectly via its lipid scramblase activity to remodel the ER is yet to be determined. On one hand, ATG9A has 3 putative disordered regions (www.uniprot.org/uniprotkb/Q7Z3C6/entry), thought to be important for biomolecular condensate formation [61], which could hint that it has a direct role in phase separation. One of these regions is actually very large in the N-terminus. On the other hand, being a scramblase, ATG9A may directly affect the membrane curvature tensions on both sides of the lipidic bilayer. So if some external entity tries to deform a membrane (in our case, it could be the vRNPs), a scramblase could help diminish the effort to induce curvature [62].

Understanding the mechanisms controlling the material properties of viral inclusions formed during IAV infection may provide new means to prevent IAV genome assembly. Future directions should involve identifying the interacting partners of ATG9A (as ATG2A has been excluded as such in our system) and the signaling pathways that promote phase separation of viral inclusions at the ER. The unveiled key biological processes may have extended relevance to other severe viral infections, which involve ER remodeling and phase separation, for example, hepatitis C virus and SARS-CoV-2.

## Methods

### Cells and viruses

The epithelial cells Madin-Darby canine kidney (MDCK) and alveolar basal (A549) were a kind gift of Prof Paul Digard, Roslin Institute, United Kingdom. The GFP-Rab11a WT$^{low}$ and DN$^{low}$ stable cell lines (in A549 background) were previously produced and characterized by us [4,6]. All cell types were cultured in Dulbecco's Modified Eagle's Medium (DMEM, Gibco, 21969035) supplemented with 10% fetal bovine serum (FBS, Gibco, 10500064), 1% penicillin/streptomycin solution (Biowest, L0022) and 2 mM L-glutamine (Gibco). GFP-Rab11 WTlow and DNlow cell culture media was also supplemented with 1.25 μg.mL$^{-1}$ puromycin (Calbiochem). Cells were regularly tested for mycoplasma contamination with the LookOut mycoplasma PCR detection kit (Sigma, MP0035), using JumpStart Taq DNA Polymerase (Sigma, D9307). Reverse genetics derived A/Puerto Rico/8/34 (PR8 WT; H1N1) virus and PR8 virus expressing PA-mNeonGreen [16] were used as model viruses and titrated by plaque assay

according to reference [6]. Virus infections were performed at a multiplicity of infection (MOI) of 3 to 10. After 45 min, cells were overlaid with DMEM containing either 0.14% bovine serum albumin (for plaque assays) or DMEM supplemented with 10% FBS, 1% penicillin/ streptomycin solution, and 2 mM L-glutamine (for immunofluorescence, western blotting, and RT-qPCR). To calculate viral titers, supernatants collected from infected cells were subjected to a plaque assay on MDCK monolayers. The drug nocodazole (Sigma, 487928) was dissolved in DMSO and used at a final concentration of 10 $\mu g.mL^{-1}$ for 2 h.

## Plasmids and siRNA

Reverse genetic plasmids were contributed by Dr Ron Fouchier (Erasmus MC, the Netherlands). The GFP-tagged ATG9A plasmid was a gift from Dr Sharon Tooze (Francis Crick Institute, UK). The plasmid encoding mCherry tagged to the ER was produced in-house and characterized by us in [4]. The siRNA targeting ATG9A (mixture of 4 siRNAs, GS79065), ATG2A (mixture of 4 siRNAs, GS23130), ULK1 (1 siRNA type, SI02223270), ULK2 (mixture of 4 siRNAs, GS9706), TBC1D14 (mixture of 4 siRNAs, GS57533), and non-targeting (NT, #5091027310) were purchased from Qiagen.

## Transfections

For plasmid transfection, cells were grown to 70% confluency in 24-well plates and transfected with 250 ng of indicated plasmids using Lipofectamine LTX and Opti-MEM (both from Life Technologies), according to manufacturer's instructions. Cells were simultaneously transfected and infected or mock-infected with PR8 virus, at MOI 10, for 8 to 12 h. Specifically for GFP-ATG9A overexpression, plasmid transfection was performed 24 h before infection. For siRNA transfection, cells were grown to 50% confluency in 6-well plates the day before transfection. Cells were transfected with siRNA (100 pmol/well) using DharmaFECT (Dharmacon) for 48 h and then infected or mock-infected with PR8 at MOI 3 for 8 h.

## High-pressure freezing/freeze substitution and electron tomography

Cells grown on 3 mm aclar disks (carbon coated) were fixed using a mixture of 2% (v/v) formaldehyde and 0.2% (v/v) glutaraldehyde (Polysciences) in 0.1 M phosphate buffer, for 2 h at RT. Cells in the aclar disks were added to a 0.04-mm deep carrier filled with 1-hexadecene and frozen using a High Pressure Freezer Compact 02 (Wohlwend Engineering Switzerland). The samples were then freeze substituted at −90°C with 0.1% (w/v) uranyl acetate and 0.01% (w/v) tannic acid (EMS) in acetone for 6 h using a Leica EM AFS2 with a processor Leica EM FSP. The temperature was then raised to −45°C at a slope of 5°C/h. Samples were stabilized at −45°C for 1.5 h before washing in acetone 3 times. Samples were infiltrated and embedded in Lowicryl HM20 (Polysciences) at −45°C. Polymerization of the resin was done using UV light at −25°C for 48 h. Sections of 120 nm (Leica UC7) were picked on palladium-copper grids coated with 1% (w/v) formvar (Agar Scientific) in chloroform (VWR). The post-staining was made with 1% (w/v) uranyl acetate and Reynolds lead citrate, for 5 min each. For tomography, 15 nm protein A-gold (UMC, Utrecht) was added to both sides of the sections before staining, as fiducial markers. Tilt-series were acquired on a FEI Tecnai G2 Spirit BioTWIN operating at 120 keV equipped with an Olympus-SIS Veleta CCD Camera. The images were aligned based on the fiducial markers, and tomograms were reconstructed and joined with the IMOD toolbox [63]. Manual segmentation of cell organelles was used to generate 3D surface models using the AMIRA software (Thermo Scientific).

## Tokuyasu—Double immunogold labeling

Cells were fixed in suspension using 2% (v/v) formaldehyde (EMS) and 0.2% (v/v) glutaraldehyde (Polysciences) in 0.1 M phosphate buffer (PB), for 2 h at RT. Subsequently, cells were centrifuged and washed with PB. The aldehydes were quenched using 0.15% (w/v) glycine (VWR) in 0.1 M PB for 10 min at RT. Cells were infiltrated in 12% (w/v) gelatin (Royal) for 30 min at 37°C and centrifuged. The gelatin was solidified on ice, cut into 1 mm$^3$ cubes, and placed in 2.3 M sucrose (Alfa Aesar) in 0.1 M PB, overnight at 4°C. The cubes were mounted onto specimen holders and frozen at −196°C by immersion into liquid nitrogen. Samples were trimmed and cut into 50-nm-thick sections (in a Leica EM-FC7 at −110°C) and laid onto formvar carbon-coated 100-mesh grids. For immunogold labeling, sections were blocked with PBS/1% BSA for 20 min at RT. Antibody staining was done sequentially in PBS/1% BSA at RT: rabbit anti-GFP (1:500, 1 h, Abcam, 6556), goat anti-rabbit IgG conjugated to 18 nm gold (1:20, 30 min; Jackson ImmunoResearch Laboratories, 111-215-144), mouse anti-NP (1:200, 1 h, Abcam, 20343), and goat anti-mouse IgG conjugated with 6 nm gold (1:20, 30 min; Jackson ImmunoResearch Laboratories, 115-195-146). Gold particles were fixed by applying 1% (v/v) formaldehyde in PBS for 5 min at RT. Blocking and extensive washing were performed in-between stainings. In the final step, gold particles were fixed using 1% (v/v) glutaraldehyde (Polysciences) for 5 min at RT. Grids were washed in distilled H$_2$O and counterstained using methyl-cellulose–uranyl acetate solution for 5 min on ice. EM images were acquired on a Hitachi H-7650 operating at 100 keV equipped with a XR41M mid-mount AMT digital camera. Images were postprocessed using Adobe Photoshop CS2 and ImageJ (NIH).

## Fixed-cell imaging

For immunofluorescence, cells were fixed for 15 min with 4% formaldehyde and permeabilized for 7 min with 0.2% (v/v) Triton-X-100 in PBS. Cells were incubated with the indicated primary antibodies for 1 h at RT, washed and incubated for 30 min with Alexa Fluor–conjugated secondary antibodies and Hoechst. Antibodies used were as follows: rabbit polyclonal against Rab11a (1:200; Proteintech, 15903-1-AP), ATG9A (1:200, Abcam, 108338), and viral NP (1:1,000; gift from Prof Paul Digard, Roslin Institute, UK); mouse monoclonal against viral NP (1:1,000; Abcam, 20343), PDI (1:500, Life Technologies, MA3-019), GM130 (1:500, BD Transduction Laboratories, 610823); mouse monoclonal against alpha-tubulin (1:1,000; b5-1-2, Sigma, T6074). Actin was stained using 100 nM of phalloidin-iFluor488 for 30 min at RT (Phalloidin-iFluor 488 Reagent, ab176753). Secondary antibodies were all from the Alexa Fluor range (1:1,000; Life Technologies). Following washing, cells were mounted with Dako Faramount Aqueous Mounting Medium, and single optical sections were acquired on either a Leica SP5 live confocal or a Zeiss LSM 980 AiryScan2 system.

## Live-cell imaging

Cells ($2 \times 10^4$/well) were grown in chambered glass-bottomed dishes (Lab-Tek) and maintained at 37°C, 5% CO$_2$ in Opti-MEM medium (Gibco) during imaging. Samples were imaged using either a Roper Spinning Disk confocal (Yokogawa CSU-X1) or a Zeiss LSM 980 AiryScan2 systems and postprocessed using Adobe Photoshop CS2 and ImageJ (NIH). For microtubule staining in live cells, 200 nM of Sir-Tubulin (Cytoskeleton) was added 10 h before imaging.

## Quantitative image analysis

**Circularity and roundness:** For shape quantifications of viral inclusions in confocal optical sections of fixed samples, we first segmented the periphery (using Rab11a or NP staining) of

the cell and the nucleus (Hoechst staining), then the nucleus was removed, followed by segmentation of the cytoplasmic viral inclusions (using Rab11a or NP staining) using a custom-made macro and ImageJ [64]. Briefly, the background of images was subtracted, thresholds adjusted automatically, and "shape descriptor" function used to determine the roundness/circularity of each viral inclusion inside selected cells. Frequency distributions were calculated and plotted with GraphPad Prism using intervals of circularity/roundness values between [0–1]. Images were postprocessed using Adobe Photoshop CS2 and ImageJ. **Mean squared displacement:** Trackmate plugin [65] was used to track the XY trajectories of viral inclusions for 10 min at a timescale of 2 s/frame in live cells, and their MSD was subsequently analyzed with a custom R (version 4.1.0) script, using the formula MSD = $(x(t) − x(0))^2 + (y(t) − y(0))^2$. **Manders' colocalization:** Before colocalization analysis, we performed noise reduction of spinning disk confocal images of live-cell imaging using an implementation of noise2void [66] in Zero-CostDL4mic [67]. Since intensities are not guaranteed to be linear after this enhancement, and we were mainly interested in testing overlap (or co-occurrence) and not correlation of fluorescence intensities between microtubules (tubulin) and viral inclusions (Rab11a or PA-mNeon-Green), we opted for Manders' co-occurrence analysis. We calculated the thresholded Manders' Overlap Coefficient (tM1) for microtubules as a measurement of association of viral inclusions with microtubules, which ensures that tM1 is irrespective of the size or shape of the viral inclusions, which would result in a biased estimation of co-occurrence. This was done at every 5 frames for each movie (10 min at a timescale of 2 s/frame). Colocalization analysis was performed in ImageJ using the "coloc2" plugin, and threshold values calculated automatically using Costes method [68] to avoid user bias. **Pearson's colocalization:** Colocalization analysis in single sections of fixed cells to establish correlation of fluorescence intensities between structures (Rab11a versus NP; ATG9A versus GM130) was performed in ImageJ using the "colocalization threshold" plugin.

## Western blotting

Western blotting was performed according to standard procedures and imaged using a LI-COR Biosciences Odyssey near-infrared platform or the Amersham chemiluminescence system. Antibodies used included the following: rabbit polyclonal against virus NP (1:1,000; a kind gift by Prof. Paul Digard, Roslin Institute, UK); rabbit polyclonal against ATG9A (1:500, Abcam, 108338); rabbit polyclonal against ATG2A (1:500, Proteintech, 23226-1-AP); mouse polyclonal against actin (1:1,000, Sigma, A5441); rabbit polyclonal against lamin B (1:1,000, Abcam, 16048); rabbit polyclonal against LC3 (1:1,000, MBL, PM0369). The secondary antibodies used were either from the IRDye range (1:10,000; LI-COR Biosciences) or linked to HRP (1:5,000; Cell Signaling Technologies, 1677074S). The original uncropped blots are included in S1 Raw Images.

## Fluorescence-activated cell sorting

Cell monolayers ($6 \times 10^6$) were trypsinized for 7 min at 37˚C, centrifuged at 1,500 rpm for 5 min, and cell pellets were resuspended in PBS containing 2% FBS. Approximately $1 \times 10^6$ cells/well were incubated for 30 min on ice with either PBS or with a monoclonal mouse antibody against viral proteins HA (neat, clone 6F6, produced in-house), NA (neat, clone 7D8, produced in-house), and M2 (1:400, clone 14C2, Abcam, 5416). Cells were then washed with PBS/2% FBS and centrifuged at 1,500 rpm for 5 min for 3 consecutive rounds. Cells were then either incubated with PBS or with a secondary antibody against mouse IgG conjugated to Alexa 568 (1:1,000, Life Technologies). Several steps of washing and centrifugation were performed to remove the unbound antibody. Upon washing with PBS/2% FBS, cells were fixed

with 2% paraformaldehyde (2% PFA) at RT for 15 min, washed again in PBS, and analyzed in a BD Fortessa X-20 flow cytometer equipped with 4 lasers 405 nm, 488 nm, 561 nm, and 640 nm; an SSC detector; and 16 detectors 6V, 2A, 5A-V, and 3V.

## Transferrin recycling assay

Cells ($5 \times 10^5$) were serum starved in FBS-free DMEM for 2 h and then were incubated with 1 μg.mL$^{-1}$ of Tf conjugated to Alexa Fluor 647 (Tf-647, Life Technologies, T-23366) for 10 min at 37°C, 5% $CO_2$. Cells used to determine the level of Tf-647 internalization were quickly washed in PBS and detached with 1 mM EDTA before fixing in 2% PFA for 10 min. Cells used to quantify Tf-647 recycling were washed in PBS and then incubated for 5, 10, or 15 min in live cell imaging solution (LCIS, in-lab adaption of Thermo Fisher Scientific LCIS), washed in PBS, detached in 1 mM EDTA, and fixed in 2% PFA for 10 min. Fixed cells were then washed in PBS and filtered before analysis using the flow cytometer LSR Fortessa X20 (BD). Median fluorescence intensities of Tf-647 were measured and analyzed using the softwares FlowJo, GraphPad Prism, and a custom-made R script.

## Quantitative real-time reverse transcription PCR (RT-qPCR)

Extraction of RNA from samples in NZYol (NZYtech, MB18501) was achieved by using the Direct-zol RNA minipreps (Zymo Research, R2052). Reverse transcription (RT) was performed using the NZY first strand cDNA synthesis kit (NZYTech, MB12502). Real-time RT-PCR was prepared in 384-well, white, thin walled plates 384-well PCR Plate (ABgene 12164142) by using SYBR Green Supermix (Biorad, 172–5124), 10% (v/v) of cDNA and 0.4 μM of each primer. The reaction was performed on a ABI QuantStudio-384 machine (Applied Biosciences), under the following PCR conditions: Cycle 1 (1 repeat): 95°C for 2 min; Cycle 2 (40 repeats): 95°C for 5 s and 60°C for 30 s; Cycle 3: 95°C for 5 s and melt curve 65°C to 95°C (increment 0.05°C each 5 s). Standard curves were prepared by serially diluting 1:5 a mock-infected sample from each experiment. Data were analyzed using the QuantStudio 7 software (Applied Biosciences). The mRNA level of host factors was quantified relative to reference GAPDH mRNA level. Expression was normalized to mock-infected cells treated with control siRNA. Primer sequences used for real-time RT-qPCR are listed in S1 Data File (oligonucleotide sheet).

## In vitro synthesis of vRNA standards

The strategy used in this study was published by [69]. The primers used to create templates containing a T7 phage promoter (TAATACGACTCACTATAGGG) sequence are listed in S1 Data File (oligonucleotide sheet). Viral gene sequences in pPolI plasmids for all PR8 segments were amplified by PCR using corresponding primer pairs and were purified using ZYMO Research DNA cleaner and Concentrator-5 (ZYMO, D4014). Purified PCR products were in vitro transcribed using the T7 RiboMAX Express Large Scale RNA Production System (Promega, P1320). The transcripts were purified using the RNeasy Micro kit (QIAGEN, 74004). The concentration of purified RNA was determined by spectrophotometry. The molecular copies of synthetic RNA were calculated based on the total molecular weight of the segment.

## RNA extraction from virions

Supernatants from virus-infected cells were centrifuged at 6,800*g* for 3 min to clear cryoprecipitates. Virion RNA was extracted using the QIAamp Viral RNA Mini kit (Qiagen, 52906)

according to manufacturer's instructions. The concentration of purified RNA was determined by spectrophotometry.

### Hot start reverse transcription with a tagged primer

cDNAs complementary to vRNA (standards and RNA isolated from virions or infected cells) were synthesized with tagged primers to add an 18- to 20-nucleotide tag at the 5′ end that was unrelated to influenza virus (vRNAtag, GGCCGTCATGGTGGCGAAT). RT with the tagged primer was performed with the hot start method modification of using saturated trehalose, as described in [69].

### vRNA-to-PFU ratio quantification

Absolute quantification of vRNA levels in isolated virions was done by real-time RT-qPCR as described above. Standard curves were generated by 100-fold serial dilutions of synthetic viral RNA. Data were analyzed using the QuantStudio software (Applied Biosciences). Primer sequences used for RT and for real-time RT-qPCR are listed in S1 Data File (oligonucleotide sheet). The ratio of vRNA levels to PFUs (vRNA-to-PFU) was calculated by dividing vRNA levels in isolated virions by the PFUs obtained from the same cell supernatants.

### vRNA-to-GAPDH relative expression

Quantification of vRNA levels in the cell cytosol was done by real-time RT-qPCR, having performed RT using the hot start method described above. Standard curves were prepared by serially diluting 1:5 an infected sample from each experiment. Data were analyzed using the QuantStudio 7 software (Applied Biosciences). The vRNA level of viral segments was quantified relative to reference GAPDH mRNA level. Expression was normalized to infected cells treated with control siRNA (siNT). Primer sequences used for real-time RT-qPCR are listed in S1 Data File (oligonucleotide sheet).

### Data quantification and statistical analysis

Data were analyzed in Prism 6 version (GraphPad Software), in R (version 4.1.0), QuantStudio 7 software (Applied Biosciences), Adobe Photoshop CS2 version, AMIRA software (Thermo Fisher Scientific), and FlowJo. Experimental replicates and tests applied to determine statistical significance between different conditions are described in each figure legend.

### Supporting information

**S1 Fig. Functionally active Rab11a is critical for montage of viral inclusions and for viral production. (A)** Cells (GFP-Rab11a WT$^{low}$ and GFP-Rab11a DN$^{low}$) were infected or mock-infected with PR8 virus for 12 h, at an MOI of 3. Viral production was determined at 12 h post-infection by plaque assay and plotted as PFUs per milliliter (mL) SEM. Data are a pool from 4 independent experiments. Statistical analysis was done by Mann–Whitney test ($^*p < 0.05$). **(B)** Formation of viral inclusions was analyzed by immunofluorescence at 12 h postinfection, using GFP-Rab11a as a proxy to detect viral inclusions. The ER distribution was detected by antibody staining against host PDI (gray). Infected cells expressing Rab11a WT were able to mount viral inclusions, whereas infected cells expressing Rab11a DN were not. Cells were stained for ER (magenta) as a cellular reference. Bar = 10 μm. ER, endoplasmic reticulum MOI, multiplicity of infection; PFU, plaque-forming unit; SEM, standard error of the mean; WT, wild-type.
(TIF)

**S2 Fig. IAV inclusions form in A549-infected cells, but not in cells expressing a functionally inactive Rab11a. (A)** Cells (A549 and GFP-Rab11a DN$^{low}$) were infected with PR8 virus for 12 h, at an MOI of 3. Cells were processed by high-pressure freezing/freeze substitution and imaged by ET-TEM. In each case, 4 sequential tomograms (of 120 nm each approximately) were acquired and stitched together. Representative cells are shown with individual sections (including section height in nm) and the 3D cumulative model. For each condition, at least 10 cells were analyzed. Bar = 500 nm. Images were extracted from S7, S8, S9, and S10 Videos. Abbreviations: pm, plasma membrane; er, endoplasmic reticulum; v, budding virions; m, mitochondria; smv, single-membrane vesicle; dmv, double-membrane vesicle. **(B)** Cells (GFP-Rab11a WT$^{low}$) were infected or mock-infected with PR8 virus for 12 h at an MOI of 3. Sections (70 nm) were stained by Tokuyasu double immunogold labeling using antibodies against GFP (18 nm-gold particle to detect Rab11a) and viral NP protein (10 nm-gold particle to detect vRNPs) and imaged by TEM. Green arrowheads show single-membrane vesicles (smv), yellow arrowheads highlight double-membrane vesicles (dmv), and blue arrowheads point to the ER (er). The yellow dashed line delimits a viral inclusion. Bar = 100 nm. ER, endoplasmic reticulum; ET, electron tomography; IAV, influenza A virus; MOI, multiplicity of infection; NP, nucleoprotein; TEM, transmission electron microscopy; vRNP, viral ribonucleoprotein; 3D, 3-dimensional.
(TIF)

**S3 Fig. Effect of depleting autophagy factors on mock-infected cells.** Cells (A549) were treated with siRNA non-targeting (siNT) or targeting ULK1 (siULK1), ULK2 (siULK2), TBC1D14 (siTBC1D14), ATG2A (siATG2A), and ATG9A (siATG9A) for 48 h and then mock-infected or infected with PR8 virus, at an MOI of 3, for 8 h. **(A)** Mock-infected cells were fixed and analyzed by immunofluorescence using an antibody against viral NP protein (magenta), host Rab11 (gray), and tubulin (green). Representative infected cells are shown in the main text in Fig 3D. **(B)** In PR8 infected cells, the correlation between roundness and circularity of viral inclusions, as marked by viral NP protein, were calculated for each condition using the Shape Descriptor tool (Image J, NIH) and were plotted as the percentage of a binned frequency distribution as shown. The maximum value of roundness and circularity (1) corresponds to a circular structure, whereas the minimum value represents a linear structure (0). Approximately 15 cells, from 2 independent experiments, were analyzed per condition. Statistical analysis was done by two-way ANOVA, followed by a Sidak's multiple comparisons test (no statistical differences were found). Circularity and roundness values for the ATG9A depleted condition are shown in the main text in Fig 3F.
(TIF)

**S4 Fig. ATG9A depletion elongates viral inclusions.** Cells (A549) were treated with siRNA non-targeting (siNT) or targeting ATG9A (siATG9A) for 48 h and then infected, at an MOI of 3, with PR8 virus for 8 h. Cells were fixed and analyzed by immunofluorescence using an antibody against viral NP protein **(A, B)** or host Rab11 **(C, D)**. The **(A, C)** roundness and **(B, D)** circularity of viral inclusions were calculated for each condition using the Shape Descriptor tool (Image J, NIH) and were plotted as the percentage of a binned frequency distribution as shown. The maximum value of roundness and circularity (1) corresponds to a circular structure, whereas the minimum value represents a linear structure (0). More than 80 cells, from 3 independent experiments, were analyzed per condition. Statistical analysis was done by two-way ANOVA, followed by a Sidak's multiple comparisons test ($^{***}p < 0.001$, $^{**}p < 0.01$).
(TIF)

**S5 Fig. ATG9A distribution is altered during IAV infection. (A)** Cells (A549) were treated with siRNA non-targeting (siNT) or targeting ATG9A (siATG9A) for 48 h and then mock-infected or infected, at an MOI of 3, with PR8 virus for 8 h. Cells were fixed and analyzed by immunofluorescence using an antibody against ATG9A (green), viral NP protein (magenta), or host GM130 (gray). Viral inclusions/vRNPs are highlighted by white boxes. Cell periphery and nuclei (blue, Hoechst staining) are delineated by yellow and white dashed lines, respectively. Bar = 10 μm. **(B, C)** Cells (A549) were transfected with a plasmid encoding GFP (as control for GFP-ATG9A) for 24 h and then infected or mock-infected with PR8 virus, at an MOI of 10, for 8 h. The localization of endogenous host proteins (GM130—Golgi or Calnexin—ER) and viral protein NP was determined by immunofluorescence using antibodies against these proteins. Nuclei (blue or gray, Hoechst staining) and cell periphery are delimited by white and yellow dashed lines, respectively. Bar = 10 μm. ER, endoplasmic reticulum; GFP, green fluorescent protein; IAV, influenza A virus; MOI, multiplicity of infection; NP, nucleoprotein; vRNP, viral ribonucleoprotein.
(TIF)

**S6 Fig. ATG9A affects viral inclusion movement on microtubules, but not on the actin cytoskeleton. (A, B)** Cells (A549) were treated with siRNA non-targeting (siNT) or targeting ATG9A (siATG9A) for 48 h and then infected, at an MOI of 3, with PR8 virus for 8 h. The localization of tubulin (green) and viral protein NP (magenta) was determined by immunofluorescence using antibodies against these proteins. The distribution of actin filaments was determined by staining with the dye phalloidin Alexa Fluor 488 (green). Nuclei (blue, Hoechst staining) and cell periphery are delimited by white and yellow dashed lines, respectively. Viral inclusions are highlighted in white boxes. Bar = 10 μm.
(TIF)

**S7 Fig. ATG9A depletion increases affinity of viral inclusions on microtubules. (A, B)** Cells (A549) were treated with siRNA non-targeting (siNT) or targeting ATG9A (siATG9A) for 48 h. Upon this period, cells were infected or mock-infected with PA-mNeonGreen PR8 virus (PAmNG.PR8, magenta) for 8 h, at an MOI of 10, and simultaneously treated with 200 nM Sir-Tubulin dye to stain the microtubules (green) in live cells. Cells were imaged for 2 min (2 s/frame) under time-lapse conditions at 8 h postinfection. White boxes show viral inclusions/PA-mNeonGreen. Individual frames with single moving particles highlighted with yellow arrows are shown in the small panels. Bar = 10 μm. Images from selected infected cells were extracted from S15 and S16 Videos. **(C, D)** Cells (A549) were treated as explained above (in **A, B**). At 8 h postinfection, cells were treated with DMSO or 10 μg/mL of nocodazole for 2 h. Cells were imaged at 10 h postinfection. White boxes show viral inclusions/PA-mNeonGreen. Bar = 10 μm. Images from selected infected cells were extracted from S17 and S18 Videos. **(E)** Each viral inclusion in a cell was tracked using the TrackMate plugin (FIJI, NIH) and displacement was quantified as explained in Fig 6G. Data were plotted as the mean squared displacement (MSD, μm$^2$) per treatment. The red dot indicates the median in the boxplots. Statistical analysis was done by a Kruskal–Wallis test (***$p < 0.001$). **(F)** Colocalization between microtubules (tubulin) and viral inclusions (PA-mNeonGreen) was determined as the Manders' Overlap Coefficient tM1 (thresholded; explained in Methods section). Between 9 and 13 cells per condition were analyzed. Statistical analysis was done by a Student $t$ test (***$p < 0.001$). Experiments were performed twice.
(TIF)

**S1 Raw Images. Uncropped western blots.** Uncropped original blots used in this study are shown. Proteins were detected using the Odyssey infrared (IR) system (green channel, 800

nm; red channel, 780 nm) or chemiluminescence Amersham system. In some cases, original blots were converted to grayscale using ImageJ (FIJI, NIH) and the brightness and contrast (B&C) was adjusted as shown for each case. The final crops shown can be found in the main text in Figs 3C, 3G, 4C and 5B, respectively. The molecular weights of the ladder in kiloDaltons (kDa) are shown for each blot.
(PDF)

**S1 Data. Table containing the raw values for all the experiments and also containing a list of the oligonucleotides used.** The different information is separated by different sheets at the bottom of the excel file.
(XLSX)

**S1 Video. Cells (GFP-Rab11a WT<sup>low</sup>, green) were simultaneously transfected with a plasmid encoding mCherry tagged to the ER (magenta) and infected with PR8 virus for 12 h at an MOI of 10.** Cells were imaged under time-lapse conditions (2 s/frame) at 12 h postinfection, using a Zeiss LSM 980 AiryScan2 system. The video shows fusion/fission events of viral inclusions (green), as well as their interaction with the ER (magenta). Bar = 10 μm. The video is part of Fig 1D.
(AVI)

**S2 Video. Cells (GFP-Rab11a WT<sup>low</sup>, green) were simultaneously transfected with a plasmid encoding mCherry tagged to the ER (magenta) and mock-infected with PR8 virus for 12 h at an MOI of 10.** Cells were imaged under time-lapse conditions (2 s/frame) at 12 h postinfection, using a Zeiss LSM 980 AiryScan2 system. The video shows fusion/fission events of viral inclusions (green), as well as their interaction with the ER (magenta). Bar = 10 μm. The video is part of Fig 1D.
(AVI)

**S3 Video. Cells (GFP-Rab11a WT<sup>low</sup>) were infected with PR8 virus for 12 h at an MOI of 3.** Cells were processed by high-pressure freezing/freeze substitution and imaged by ET-TEM. For the 3D model reconstruction, 4 serial tomograms of 120 nm each were stitched together (480 nm thickness in total). Cellular structures were manually segmented and artificially colored, as follows: plasma membrane, gray; ER, blue; budding virions, pink; mitochondria, purple; single-membrane vesicle, light green; double-membrane vesicle, yellow; ER dilation, dark green. The video is part of Fig 2D. ER, endoplasmic reticulum; ET, electron tomography; MOI, multiplicity of infection; TEM, transmission electron microscopy; 3D, 3-dimensional.
(AVI)

**S4 Video. Cells (GFP-Rab11a WT<sup>low</sup>) were infected with PR8 virus for 12 h at an MOI of 3.** Cells were processed by high-pressure freezing/freeze substitution and imaged by ET-TEM. Four serial tomograms of 120 nm each were stitched together (480 nm thickness in total). Original tomogram join is shown without segmentation of cellular structures. The video is part of Fig 2D. ET, electron tomography; MOI, multiplicity of infection; TEM, transmission electron microscopy.
(AVI)

**S5 Video. Cells (GFP-Rab11a WT<sup>low</sup>) were mock-infected with PR8 virus for 12 h at an MOI of 3.** Cells were processed by high-pressure freezing/freeze substitution and imaged by ET-TEM. For the 3D model reconstruction, 4 serial tomograms of 120 nm each were stitched together (480 nm thickness in total). Cellular structures were manually segmented and artificially colored, as follows: plasma membrane, gray; ER, blue; budding virions, pink; mitochondria, purple; single-membrane vesicle, light green; double-membrane vesicle, yellow; ER

dilation, dark green. The video is part of Fig 2C. ER, endoplasmic reticulum; ET, electron tomography; MOI, multiplicity of infection; TEM, transmission electron microscopy; 3D, 3-dimensional.
(AVI)

**S6 Video. Cells (GFP-Rab11a WT^low) were mock-infected with PR8 virus for 12 h at an MOI of 3.** Cells were processed by high-pressure freezing/freeze substitution and imaged by ET-TEM. Four serial tomograms of 120 nm each were stitched together (480 nm thickness in total). Original tomogram join is shown without segmentation of cellular structures. The video is part of Fig 2C. ET, electron tomography; MOI, multiplicity of infection; TEM, transmission electron microscopy.
(AVI)

**S7 Video. Cells (A549) were infected with PR8 virus for 12 h, at an MOI of 3.** Cells were processed by high-pressure freezing/freeze substitution and imaged by ET-TEM. For the 3D model reconstruction, 4 serial tomograms of 120 nm each were stitched together (480 nm thickness in total). Cellular structures were manually segmented and artificially colored, as follows: plasma membrane, gray; ER, blue; budding virions, pink; mitochondria, purple; single-membrane vesicle, light green; double-membrane vesicle, yellow; ER dilation, dark green. The video is part of S2A Fig. ER, endoplasmic reticulum; ET, electron tomography; MOI, multiplicity of infection; TEM, transmission electron microscopy; 3D, 3-dimensional.
(AVI)

**S8 Video. Cells (A549) were infected with PR8 virus for 12 h, at an MOI of 3.** Cells were processed by high-pressure freezing/freeze substitution and imaged by ET-TEM. Four serial tomograms of 120 nm each were stitched together (480 nm thickness in total). Original tomogram join is shown without segmentation of cellular structures. The video is part of S2A Fig. ET, electron tomography; MOI, multiplicity of infection; TEM, transmission electron microscopy.
(AVI)

**S9 Video. Cells (GFP-Rab11a DN^low) were infected with PR8 virus for 12 h, at an MOI of 3.** Cells were processed by high-pressure freezing/freeze substitution and imaged by ET-TEM. For the 3D model reconstruction, 4 serial tomograms of 120 nm each were stitched together (480 nm thickness in total). Cellular structures were manually segmented and artificially colored, as follows: plasma membrane, gray; ER, blue; budding virions, pink; mitochondria, purple; single-membrane vesicle, light green; double-membrane vesicle, yellow; ER dilation, dark green. The video is part of S2A Fig. ER, endoplasmic reticulum; ET, electron tomography; MOI, multiplicity of infection; TEM, transmission electron microscopy; 3D, 3-dimensional.
(AVI)

**S10 Video. Cells (GFP-Rab11a DN^low) were infected with PR8 virus for 12 h, at an MOI of 3.** Cells were processed by high-pressure freezing/freeze substitution and imaged by ET-TEM. Four serial tomograms of 120 nm each were stitched together (480 nm thickness in total). Original tomogram join is shown without segmentation of cellular structures. The video is part of S2A Fig. ET, electron tomography; MOI, multiplicity of infection; TEM, transmission electron microscopy.
(AVI)

**S11 Video. Cells (GFP-Rab11a WT^low, magenta) were treated with siRNA non-targeting (siNT) for 48 h.** Upon this period, cells were infected with PR8 virus for 8 h, at an MOI of 3, and simultaneously treated with 200 nM Sir-Tubulin dye to stain the microtubules (green) in

live cells. Cells were imaged for 10 min (2 s/frame) under time-lapse conditions at 8 h postinfection, using a Roper Spinning Disk confocal (Yokogawa CSU-X1). The video shows the dynamics of viral inclusions/Rab11a (magenta) and microtubules (green). Bar = 10 μm. The video is part of Fig 6A.
(AVI)

**S12 Video. Cells (GFP-Rab11a WT<sup>low</sup>, magenta) were treated with siRNA targeting ATG9A (siATG9A) for 48 h.** Upon this period, cells were infected with PR8 virus for 8 h, at an MOI of 3, and simultaneously treated with 200 nM Sir-Tubulin dye to stain the microtubules (green) in live cells. Cells were imaged for 10 min (2 s/frame) under time-lapse conditions at 8 h postinfection, using a Roper Spinning Disk confocal (Yokogawa CSU-X1). The video shows the dynamics of viral inclusions/Rab11a (magenta) and microtubules (green). Bar = 10 μm. The video is part of Fig 6B.
(AVI)

**S13 Video. Cells (GFP-Rab11a WT<sup>low</sup>, magenta) were treated with siRNA non-targeting (siNT) for 48 h.** Upon this period, cells were mock-infected with PR8 virus for 8 h, at an MOI of 3, and simultaneously treated with 200 nM Sir-Tubulin dye to stain the microtubules (green) in live cells. Cells were imaged for 10 min (2 s/frame) under time-lapse conditions at 8 h post infection, using a Roper Spinning Disk confocal (Yokogawa CSU-X1). The video shows the dynamics of viral inclusions/Rab11a (magenta) and microtubules (green). Bar = 10 μm. The video is part of Fig 6C.
(AVI)

**S14 Video. Cells (GFP-Rab11a WT<sup>low</sup>, magenta) were treated with siRNA targeting ATG9A (siATG9A) for 48 h.** Upon this period, cells were mock-infected with PR8 virus for 8 h, at an MOI of 3, and simultaneously treated with 200 nM Sir-Tubulin dye to stain the microtubules (green) in live cells. Cells were imaged for 10 min (2 s/frame) under time-lapse conditions at 8 h postinfection, using a Roper Spinning Disk confocal (Yokogawa CSU-X1). The video shows the dynamics of viral inclusions/Rab11a (magenta) and microtubules (green). Bar = 10 μm. The video is part of Fig 6D.
(AVI)

**S15 Video. Cells (A549) were treated with siRNA non-targeting (siNT) for 48 h.** Upon this period, cells were infected with PA-mNeonGreen PR8 virus (PAmNG.PR8, magenta) for 8 h, at an MOI of 10, and simultaneously treated with 200 nM Sir-Tubulin dye to stain the microtubules (green) in live cells. At 8 h postinfection, cells were treated with DMSO for 2 h. Cells were imaged under time-lapse conditions for 2 min (2 s/frame) at 10 h postinfection, using a Roper Spinning Disk confocal (Yokogawa CSU-X1). The video shows the dynamics of viral inclusions/PA-mNeonGreen (magenta) and microtubules (green). Bar = 10 μm. The video is part of S7A Fig.
(AVI)

**S16 Video. Cells (A549) were treated with siRNA targeting ATG9A (siATG9A) for 48 h.** Upon this period, cells were infected with PA-mNeonGreen PR8 virus (PAmNG.PR8, magenta) for 8 h, at an MOI of 10, and simultaneously treated with 200 nM Sir-Tubulin dye to stain the microtubules (green) in live cells. At 8 h postinfection, cells were treated with DMSO for 2 h. Cells were imaged under time-lapse conditions for 2 min (2 s/frame) at 10 h postinfection, using a Roper Spinning Disk confocal (Yokogawa CSU-X1). The video shows the dynamics of viral inclusions/PA-mNeonGreen (magenta) and microtubules (green). Bar = 10 μm.

The video is part of S7B Fig.
(AVI)

**S17 Video. Cells (A549) were treated with siRNA non-targeting (siNT) for 48 h.** Upon this period, cells were infected with PA-mNeonGreen PR8 virus (PAmNG.PR8, magenta) for 8 h, at an MOI of 10, and simultaneously treated with 200 nM Sir-Tubulin dye to stain the microtubules (green) in live cells. At 8 h postinfection, cells were treated with 10 μg/mL of nocodazole for 2 h. Cells were imaged under time-lapse conditions for 2 min (2 s/frame) at 10 h postinfection, using a Roper Spinning Disk confocal (Yokogawa CSU-X1). The video shows the dynamics of viral inclusions/PA-mNeonGreen (magenta) and microtubules (green). Bar = 10 μm. The video is part of S7C Fig.
(AVI)

**S18 Video. Cells (A549) were treated with siRNA targeting ATG9A (siATG9A) for 48 h.** Upon this period, cells were infected with PA-mNeonGreen PR8 virus (PAmNG.PR8, magenta) for 8 h, at an MOI of 10, and simultaneously treated with 200 nM Sir-Tubulin dye to stain the microtubules (green) in live cells. At 8 h postinfection, cells were treated with 10 μg/mL of nocodazole for 2 h. Cells were imaged under time-lapse conditions for 2 min (2 s/frame) at 10 h postinfection, using a Roper Spinning Disk confocal (Yokogawa CSU-X1). The video shows the dynamics of viral inclusions/PA-mNeonGreen (magenta) and microtubules (green). Bar = 10 μm. The video is part of S7D Fig.
(AVI)

## Acknowledgments

The authors acknowledge Prof Paul Digard (Roslin Institute, UK), Dr Ron Fouchier (Erasmus MC, the Netherlands), and Dr Sharon Tooze (Francis Crick Institute) for providing cells and reagents. The authors thank Dr Marta Alenquer (CBR, Portugal) and Dr Caitlin Simpson (University of Bristol, UK) for critical reading of the manuscript. The authors are also grateful to the Flow Cytometry Facility, Advanced Imaging Facility, and Electron Microscopy Facility at the IGC for technical support, sample processing, and data collection.

## Author Contributions

**Conceptualization:** Sílvia Vale-Costa, Maria João Amorim.

**Formal analysis:** Sílvia Vale-Costa, Temitope Akhigbe Etibor, Daniela Brás, Ana Laura Sousa, Mariana Ferreira, Gabriel G. Martins, Victor Hugo Mello.

**Funding acquisition:** Maria João Amorim.

**Investigation:** Sílvia Vale-Costa, Temitope Akhigbe Etibor, Daniela Brás, Ana Laura Sousa, Victor Hugo Mello.

**Methodology:** Sílvia Vale-Costa, Temitope Akhigbe Etibor, Ana Laura Sousa, Gabriel G. Martins, Victor Hugo Mello.

**Project administration:** Maria João Amorim.

**Supervision:** Gabriel G. Martins, Maria João Amorim.

**Validation:** Temitope Akhigbe Etibor, Daniela Brás, Ana Laura Sousa, Mariana Ferreira, Gabriel G. Martins, Victor Hugo Mello, Maria João Amorim.

**Writing – original draft:** Sílvia Vale-Costa, Daniela Brás, Maria João Amorim.

**Writing – review & editing:** Sílvia Vale-Costa, Temitope Akhigbe Etibor, Daniela Brás, Gabriel G. Martins, Victor Hugo Mello, Maria João Amorim.

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
