## [Editor Report · Decision Letter 0]

7 Aug 2023

Dear Dr. Amorim, 

Thank you for submitting your manuscript entitled "ATG9A regulates dissociation of recycling endosomes from microtubules leading to formation of influenza A virus liquid condensates" for consideration as a Research Article by PLOS Biology.

Your manuscript has now been evaluated by the PLOS Biology editorial staff, as well as by an academic editor with relevant expertise, and I am writing to let you know that we would like to send your submission back for external peer review. We will try to engage the same reviewers that reviewed your first submission.

Once your full submission is complete, your paper will undergo a series of checks in preparation for peer review. After your manuscript has passed the checks it will be sent out for review. To provide the metadata for your submission, please Login to Editorial Manager (https://www.editorialmanager.com/pbiology) within two working days, i.e. by Aug 09 2023 11:59PM.

Kind regards,

Paula

---

Senior Editor

PLOS Biology

---

## [Decision Letter · Decision Letter 1]

23 Sep 2023

Dear Dr. Amorim,

Thank you for your patience while we considered your revised manuscript "ATG9A regulates dissociation of recycling endosomes from microtubules leading to formation of influenza A virus liquid condensates" for consideration as a Research Article at PLOS Biology. Your revised study has now been evaluated by the PLOS Biology editors, the Academic Editor and 2 of the original reviewers. 

In light of the reviews, which you will find at the end of this email, we are pleased to offer you the opportunity to address the remaining points from the reviewers in a revision that we anticipate should not take you very long. We will then assess your revised manuscript and your response to the reviewers' comments with our Academic Editor aiming to avoid further rounds of peer-review.

Please also address the following formatting and journal policy requests:

1. We suggests a change in the title: "ATG9A regulates the dissociation of recycling endosomes from microtubules to form influenza A virus liquid condensates" or "ATG9A regulates the dissociation of recycling endosomes from microtubules to form liquid influenza A virus inclusions".

2. Please provide a blurb which (if accepted) will be included in our weekly and monthly Electronic Table of Contents, sent out to readers of PLOS Biology, and may be used to promote your article in social media. The blurb should be about 30-40 words long and is subject to editorial changes. It should, without exaggeration, entice people to read your manuscript. It should not be redundant with the title and should not contain acronyms or abbreviations.

**IMPORTANT - SUBMITTING YOUR REVISION**

*Resubmission Checklist*

*Published Peer Review*

*PLOS Data Policy*

*Blot and Gel Data Policy*

Sincerely,

Paula

---

Senior Editor

PLOS Biology

REVIEWS:

Reviewer #1: Virus assembly.

Reviewer #2: Autophagy and virus infection.

Reviewer #1: An important step in Influenza A virus (IAV) biogenesis is the coalescing of new viral genome segments and viral NP protein in condensates in the cytoplasm. These condensates have previously been shown to form at ER exit sites, move along microtubules, and contain Rab11 - a small GTPase that is normally associated with recycling endosomes.

In this paper, Vale-Costa report that the autophagy protein ATG9A plays a role in condensate dynamics. ATG9A is a lipid scramblase that is thought to facilitate the growth of the phagophore membrane in autophagy, in addition to having some seemingly non-autophagic functions. The effect of ATG9A on IAV replication is not dramatic (in terms of its effect on virus release), but the cell biology is still interesting.

The core finding in this manuscript, in my opinion, is the altered appearance and dynamics of IAV condensates upon siRNA depletion of ATG9A. The condensates become drawn out, and the authors use beautiful live-cell imaging to show that this is due to increased association with microtubules.

I shall be frank and say that I reviewed a previous version of this manuscript for PLOS Biology, and even though this is formally considered a new submission, the authors have revised the manuscript according to the reviewers suggestions.

In my opinion, the new manuscript is substantially better in many ways. It is easier to follow the logics of the paper, and some crucial additional experiments tie the paper together better than before. Of my major criticisms to the previous version of the manuscript (I assume they will be published as part of the rebuttal letter), most have been dealt with satisfactorily. The only major point I made that the authors didn't address is the following: "The paper lacks mechanistic detail as to how ATG9 might have its effect on IAV condensates. The fluorescence microscopy is largely well made, but the kind of proximity of fluorescence signals shown in Fig. 3F is not a proof of anything. The authors must provide more insights into how ATG9 is involved. One example of such insights would be convincing proof of interaction between ATG9 and some viral proteins." Since that concern was not addressed, the new manuscript leaves one elephant in the room untouched: how (in terms of biochemical interactions) can ATG9A, a lipid scramblase, regulate the association of vRNP condensates to microtubules and ER? However, in the new manuscript the authors clearly discuss that this is not resolved, they put it in the context of similar phase separation phenomena, and they put forward some ideas related to regions of ATG9A. Since my other major concerns were addressed, the paper has reached a maturity that, in my opinion, warrants publication in PLOS Biology.

I still have some remaining smaller comments that I think the authors can sort out between themselves and the editor without my re-re-review:

1. Simply as a comment to editor and authors: even though this is formally a new submission, it would really have helped the reviewers to get a manuscript where changes are highlighted. It takes several hours longer for a reviewer to carefully re-assess the manuscript when this is not done.

2. "However, we find that key initial players in autophagy (ULK1/2, TBC1D14 or ATG2A) did not give rise to a similar phenotype in IAV infected cells, suggesting that this function of ATG9A is novel" I think this and some other similar statements are misleading the reader given that ULK2 had a larger effect on virion production than ATG9A. What they authors say is not directly incorrect, but it might be better to rephrase this to make clear that ULK2 has a role, but probably another one.

3. Comment to the last paragraph on page 20: The authors don't show any quantitation here. I assume that it would not work since it looks like the overexpressed ATG9A-EGFP is simply all over the cytoplasm. I think the data can be kept but the limitation of this experiment in terms of possibly different a priori distribution of ATG9 should be thoroughly mentioned.

4. Comment about the subheading "ATG9A impacts viral inclusion formation without affecting the recycling endosome". This description is imprecise and clearly overstating what the authors have shown. It should be changed to something more correct and moderate. The tile of Fig. 5 is also an overstatement since the authors have hardly experimentally verified that "Rab11a-recycling", in all the meanings of that term, is unaffected.

5. "This defect, however is unlikely to be related to the recycling endosome as ATG9A depletion did not interfere with the association of vRNPs to Rab11a vesicles." See comment above. This interpretation is too broad.

6. "We observed that in siNT-treated and infected cells, viral inclusions became larger with little motility upon nocodazole treatment" The authors should either support this statement with quantitation, or change it.

7. Comment on the paragraph "ATG9A impacts efficiency of viral genome assembly but not genome packaging into virions ": I think it would help the readers to clearly define what the authors mean by "genome assembly" and "genome packaging". These terms are widely used in virology and usually have a different meaning than here. They are often even synonymous.

8. Comment on Fig. 8: The figure and its legend would benefit from clarifications. It should be described explicitly what the two versions of the cell represent, and also what the arrow from one version to the other represents.

9. "In sum, our results demonstrate that the recycling pathway is impaired during IAV infection. Moreover, enlarged cytosolic Rab11a puncta (corresponding to the liquid viral inclusions) are detected near the ER only in infected cells expressing an active Rab11a, which agrees with our previous results [4,6,7]." The Rab11a-related statement now appears before these results (new Fig. 1D-E) are discussed in the text. That should be changed.

10. "How assembled genomes reach the budding sites at the plasma membrane is not yet known, but such a question is outside the scope of this study (Fig 1E, step 4)" Should it not be Fig. 1A?

Reviewer #2: The increased data from the silencing screen indicates specific effects of ATG9A rather than on autophagy. Overall, the text is easier to follow but some paragraphs are still difficult to understand. For example (Fig 1C, 1C) a transferrin recycling assay is used to see if IAV impairs Rab-11a regulated recycling. The conclusion below states that ''recycling occurs primarily independently of Rab11a''. So this makes it difficult to see how the assay is testing the effects of IAV on Rab11a.

''This indicates that transferrin recycling occurs primarily independently of Rab11a, through a compensatory mechanism likely mediated by other Rabs operating in the recycling pathway (Rab4, Rab10) [15] in uninfected cells but that in infection by itself reduces recycling of transferrin and dependency for Rab11a.''

The rebuttal is supplied with diagrams describing interaction of vRNA with Rab11a and formation of viral inclusions, but the manuscript focusses on the role played by ER exit sites and ATG9A in tethering inclusions to microtubules. It is not clear why ER exit sites, ATG9A and microtubules do not feature in the diagram supplied for referees. Fig 8 in the manuscript provides a more detailed diagram. Addition of numbers to the arrows would have made this easier to navigate.

This is a very long paper and would benefit from editing to cut out repetition between introduction and discussion. In this respect the Rab11a data, which takes up considerable space in terms of figures and text seems to confirm what has been published by this group and others. The data show that Rab11a and Atg9A work independently of each other so the authors might consider reducing the emphasis on Rab11a data that is in agreement with published work, and focus on ATtg9A.

Referee response to rebuttal.

The specific comments have been addressed.

1. ''The first experiments describe the use of high pressure freezing and EM tomography to provide 3D reconstructions of ER:LO interface. New double membraned vesicles, single membrane vesicles and enlarged ER lumen are seen close to the ER.''

It is not clear how this relates to LOs that do not have a limiting membrane. 

This has been clarified in the text. 

2. ''The text ln163 says 'It is noteworthy that numerous vRNPs detected inside viral inclusions were not attached to any membrane (Fig 1C). '' These are not indicated in the figure. At this point the data suggest that Rab11 is a marker for membrane compartments, rather than LO.

The use of Rab11 as a marker is now explained much better.

3. ''A line scan follows fusion of LO with the ER in infected cells but does not follow interaction of Rab11 with ER in uninfected cells. The videos are difficult to interpret. A line scan for interaction in mock infected cells should be provided to support the main conclusions that ER contacts are increased for Rab11 after infection.

New line scans have been introduced. In Fig 1E.

4. ''Silencing of ATG9A produces a small drop in viral titre and converts LO into tubular networks and appears to reduce association of LO with ER. Here the virus is tracked with antibodies to NP. Some explanation for why the experiments track NP, rather than Rab11, is needed.

The reduced signal to noise ratio generated by NP staining provides a logical explanation.

5. ''LO contacts are reduced following silencing of ATG9A. '' The quantification is difficult to follow because arrows in figure 2E and 2F indicate different events making comparison difficult. Is the conclusion based on a visual interpretation where rounded LO are easier to image than tubular LO, or it is easier to surround a round LO rather than a tube, without making contacts?

An explanation is provided and shapes ''roundness'' have been re calculated.

6. ''The paper concludes '' that ATG9A is critical for proper establishment of IAV inclusions at the ER but is unlikely to be mobilized from the recycling endosome nor does it influence the association of vRNPs to Rab11a vesicles.''' This is difficult to follow because the images do not look at the ER or the effect of loss of recycling endosomes following expression of DN Rab11.

The roles of Rab11a and ATG9A have been explained more clearly emphasising that the results show that ATG9A and Rab11a have independent functions.

7. ''Again the text refers to Rab11 vesicles which most cell biologists would think are recycling endosomes rather than liquid organelles. At other times they are called viral inclusions, but are these liquid organelles?''

These changes in terminology have been clarified.

---

## [Editor Report · Decision Letter 2]

13 Oct 2023

Dear Dr Amorim,

Thank you for the submission of your revised Research Article "ATG9A regulates the dissociation of recycling endosomes from microtubules to form liquid influenza A virus inclusions" for publication in PLOS Biology. On behalf of my colleagues and the Academic Editor, Andrew Mehle, I am pleased to say that we can in principle accept your manuscript for publication, provided you address any remaining formatting and reporting issues. These will be detailed in an email you should receive within 2-3 business days from our colleagues in the journal operations team; no action is required from you until then. Please note that we will not be able to formally accept your manuscript and schedule it for publication until you have completed any requested changes.

PRESS

Sincerely, 

Paula

---

Senior Editor

PLOS Biology
